# VQGRAPH: RETHINKING GRAPH REPRESENTATION SPACE FOR BRIDGING GNNS AND MLPS

**Ling Yang**[1]   **Ye Tian**[1]*   **Minkai Xu**[3]   **Zhongyi Liu**[2]   **Shenda Hong**[1]   **Wei Qu**[2]
**Wentao Zhang**[1]   **Bin Cui**[1†]   **Muhan Zhang**[1†]   **Jure Leskovec**[3]
[1]Peking University   [2]Ant Group   [3]Stanford University
yangling0818@163.com, tyfeld@stu.pku.edu.cn,
{zhongyi.lzy, qingze.qw}@antgroup.com,{minkai, jure}@cs.stanford.edu
{hongshenda, wentao.zhang, bin.cui, muhan}@pku.edu.cn

## ABSTRACT

GNN-to-MLP distillation aims to utilize knowledge distillation (KD) to learn computationally-efficient multi-layer perceptron (student MLP) on graph data by mimicking the output representations of teacher GNN. Existing methods mainly make the MLP to mimic the GNN predictions over a few class labels. However, the class space may not be expressive enough for covering numerous diverse local graph structures, thus limiting the performance of knowledge transfer from GNN to MLP. To address this issue, we propose to learn a new powerful graph representation space by directly labeling nodes' diverse local structures for GNN-to-MLP distillation. Specifically, we propose a variant of VQ-VAE (Van Den Oord et al., 2017) to learn a structure-aware tokenizer on graph data that can encode each node's local substructure as a discrete code. The discrete codes constitute a *codebook* as a new graph representation space that is able to identify different local graph structures of nodes with the corresponding code indices. Then, based on the learned codebook, we propose a new distillation target, namely *soft code assignments*, to directly transfer the structural knowledge of each node from GNN to MLP. The resulting framework VQGRAPH achieves new state-of-the-art performance on GNN-to-MLP distillation in both transductive and inductive settings across seven graph datasets. We show that VQGRAPH with better performance infers faster than GNNs by 828×, and also achieves accuracy improvement over GNNs and stand-alone MLPs by 3.90% and 28.05% on average, respectively. Our code is available at https://github.com/YangLing0818/VQGraph

## 1 INTRODUCTION

Graph Neural Networks (GNNs) (Yang & Hong, 2022; Li et al., 2020a; Perozzi et al., 2014; Xu et al., 2019; Morris et al., 2019; Yang et al., 2020; Chen et al., 2020b) have been widely used due to their effectiveness in dealing with non-Euclidean structured data, and have achieved remarkable performances in various graph-related tasks (Hamilton et al., 2017; Kipf & Welling, 2017; Veličković et al., 2018). Modern GNNs rely on message passing mechanism to learn node representations (Yang et al., 2020). GNNs have been especially important for recommender systems (Fan et al., 2019; He et al., 2020; Wu et al., 2022; Xiao et al., 2023; Zhang et al., 2024), fraud detection (Dou et al., 2020; Liu et al., 2021; Yang et al., 2023), and information retrieval (Li et al., 2020b; Mao et al., 2020). Numerous works (Pei et al., 2020b) focus on exploring more effective ways to leverage informative neighborhood structure for improving GNNs (Park et al., 2021; Zhu et al., 2021; Zhao et al., 2022; Tang et al., 2022; Lee et al., 2021; Chien et al., 2022; Abu-El-Haija et al., 2019).

It is challenging to scale GNNs to large-scale applications which are constrained by latency and require fast inference (Zhang et al., 2020; 2022a; Jia et al., 2020), because message passing necessitates fetching topology and features of many neighbor nodes for inference on a target node, which

---

*Contributed equally.
†Corresponding authors.

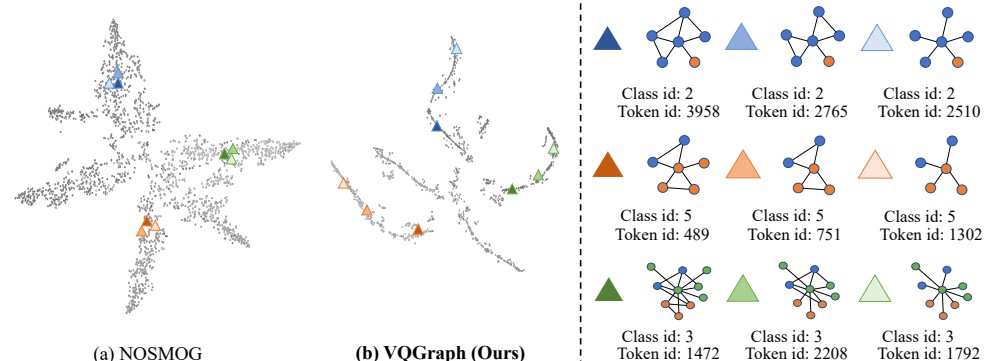

Figure 1: The t-SNE visualization of the learned graph representation space in two kinds of teacher GNNs: (a) previous SOTA "class-based" NOSMOG (Tian et al., 2023b) and (b) our "structure-based" VQGraph. "class-based" denotes learning with class labels, and "structure-based" denotes learning with our local structure reconstruction. Our learned space is more compact. We here provide both class labels and our structure labels along with illustrative substructures for demonstration.

is time-consuming and computation-intensive. Multi-layer perceptrons (MLPs) are efficient alternatives to deploy on graphs that only depend on node feature without the need of explicit message passing (Zhang et al., 2022b). Thus, recent methods use *knowledge distillation* (Hinton et al., 2015; Tian et al., 2023a; Gou et al., 2021; Yuan et al., 2020; Zhou & Song, 2021) to transfer the learned structural knowledge from GNNs to MLPs (Zhang et al., 2022b; Zheng et al., 2022; Tian et al., 2023b), which build the statistical associations between node features and class labels by making the MLP mimic the (well-trained) GNN's predictions. Then only MLPs are deployed for inference, which can also perform well in real-world graphs.

Despite some progress, current GNN-to-MLP distillation methods have a common fundamental issue: their graph representation spaces of GNN are mainly learned by a few class labels, and the class space may not be expressive enough for covering numerous diverse local graph structures of nodes, limiting the distillation performance. We explain this problem by using t-SNE (Van der Maaten & Hinton, 2008) to visualize the graph representation space as in Figure 1. We can observe that the graph representation space in previous teacher GNN is not expressive enough for identifying fine-grained local structural differences between nodes of the same class, which may limit the structural knowledge transfer from GNN to MLP.

Here we introduce a new powerful graph representation space for bridging GNNs and MLPs by directly labeling diverse nodes' local structures. Specifically, we propose a variant of VQ-VAE (Van Den Oord et al., 2017) to learn a structure-aware tokenizer on graph data that can encode each node with its substructure as a discrete code. The numerous codes constitute a *codebook* as our new graph representation space that is able to identify different local neighborhood structures of nodes with the corresponding code indices. As demonstrated in Figure 1, our learned representation space is more expressive and can identify subtle differences between nodes' local structures. Based on the learned codebook, we can effectively facilitate the structure-based distillation by maximizing the consistency of *soft code assignments* between GNN and MLP models, given by the KL divergence between GNN predictions and MLP predictions over the discrete codes of the codebook.

We highlight our main contributions as follows: **(i)** To the best of our knowledge, we for the first time directly learn to label nodes' local neighborhood structures to acquire a powerful node representation space (*i.e.*, a *codebook*) for bridging GNNs and MLPs. **(ii)** Based on the learned codebook, we utlize a new distillation target with *soft code assignments* to effectively facilitate the structure-aware knowledge distillation. We further conduct both visualization and statistical analyses for better understanding with respect to our superior local and global structure awareness for GNN-to-MLP distillation. **(iii)** Extensive experiments across seven datasets show VQGRAPH can consistently outperform GNNs by **3.90%** on average accuracy, while enjoying **828×** faster inference speed. Also VQGRAPH outperforms MLPs and SOTA distillation method NOSMOG (Tian et al., 2023b) by **28.05% and 1.39%** on average accuracy across datasets, respectively.

## 2 RELATED WORK

**Inference Acceleration for GNNs** Pruning (Zhou et al., 2021) and quantizing GNN parameters (Zhao et al., 2020) have been studied for inference acceleration (Chen et al., 2016; Judd et al.,

2016; Han et al., 2015; Gupta et al., 2015). Although these approaches accelerate GNN inference to a certain extent, they do not eliminate the neighbor-fetching latency. Graph-MLP (Hu et al., 2021) proposes to bypass GNN neighbor fetching by learning a computationally-efficient MLP model with a neighbor contrastive loss, but its paradigm is only transductive and can be not applied in the more practical inductive setting. Besides, some works try to speed up GNN in training stage from the perspective of node sampling (Zou et al., 2019; Chen et al., 2018c), which are complementary to our goal on inference acceleration.

**Knowledge Distillation for GNNs** Existing GNN-based knowledge distillation methods try to distill teacher GNNs to smaller student GNNs (GNN-GNN distillation) or MLPs (GNN-to-MLP distillation). Regarding the GNN-GNN distillation, LSP (Yang et al., 2021c), TinyGNN (Yan et al., 2020), GFKD (Deng & Zhang, 2021), and GraphSAIL (Xu et al., 2020) conduct KD by enabling student GNN to maximally preserve local information that exists in teacher GNN. The student in CPF (Yang et al., 2021b) is not a GNN, but it is still heavily graph-dependent as it uses LP (Zhu & Ghahramanih, 2002; Huang et al., 2021). Thus, these methods still require time-consuming neighbors fetching. To address these latency issues, recent works focus on GNN-to-MLP distillation that does not require message passing, as seen in (Hu et al., 2021; Zhang et al., 2022b; Zheng et al., 2022; Tian et al., 2023b). For example, recent sota methods GLNN (Zhang et al., 2022b) and NOSMOG (Tian et al., 2023b) train the student MLP with node features as inputs and class predictions from the teacher GNN as targets. However, class predictions over a few labels, as their distillation targets, can not sufficiently express structural knowledge of graph structures as discussed in Sec. 1. Hence, we for the first time propose to directly label nodes' local neighborhood structures to facilitate structure-aware knowledge distillation.

## 3 PRELIMINARIES

**Notation and Graph Neural Networks** We denote a graph as $\mathcal{G} = (\mathcal{V}, \boldsymbol{A})$, with $\mathcal{V} = \{\boldsymbol{v}_1, \boldsymbol{v}_2, \cdots, \boldsymbol{v}_n\}$ represents all nodes, and $\boldsymbol{A}$ denotes adjacency matrix, with $A_{i,j} = 1$ if node $\boldsymbol{v}_i$ and node $\boldsymbol{v}_j$ are connected, and 0 otherwise. Let $N$ denote the total number of nodes. $\boldsymbol{X} \in \mathbb{R}^{N \times D}$ represents the node feature matrix with each raw being a $D$-dimensional node attribute $\boldsymbol{v}$. For node classification, the prediction targets are $\boldsymbol{Y} \in \mathbb{R}^{N \times K}$, where row $\boldsymbol{y}_v$ is a $K$-dim one-hot vector for node $v$. For a given $\mathcal{G}$, usually a small portion of nodes will be labeled, which we mark using superscript $^L$, i.e. $\mathcal{V}^L$, $\boldsymbol{X}^L$ and $\boldsymbol{Y}^L$. The majority of nodes will be unlabeled, and we mark using the superscript $^U$, i.e. $\mathcal{V}^U$, $\boldsymbol{Y}^U$ and $\boldsymbol{Y}^U$. For a given node $\boldsymbol{v} \in \mathcal{V}$, GNNs aggregate the messages from node neighbors $\mathcal{N}(\boldsymbol{v})$ to learn node embedding $\boldsymbol{h}_{\boldsymbol{v}} \in \mathbb{R}^{d_n}$ with dimension $d_n$. Specifically, the node embedding in $l$-th layer $\boldsymbol{h}_{\boldsymbol{v}}^{(l)}$ is learned by first aggregating (AGG) the neighbor embeddings and then updating (UPDATE) it with the embedding from the previous layer. The whole learning process can be denoted as: $\boldsymbol{h}_{\boldsymbol{v}}^{(l)} = \text{UPDATE}(\boldsymbol{h}_{\boldsymbol{v}}^{(l-1)}, \text{AGG}(\{\boldsymbol{h}_{\boldsymbol{u}}^{(l-1)} : \boldsymbol{u} \in \mathcal{N}(\boldsymbol{v})\}))$.

**Vector Quantized-Variational AutoEncoder (VQ-VAE) for Continuous Data** The VQ-VAE model (Van Den Oord et al., 2017) is originally proposed for modeling continuous data distribution, such as images, audio and video. It encodes observations into a sequence of discrete latent variables, and reconstructs the observations from these discrete variables. Both encoder and decoder use a shared codebook. More formally, the encoder is a non-linear mapping from the input space, $\boldsymbol{x}$, to a vector $E(\boldsymbol{x})$. This vector is then quantized based on its distance to the prototype vectors (tokens) in the codebook $\boldsymbol{e}_k, k \in 1 \ldots K$ such that each vector $E(\boldsymbol{x})$ is replaced by the index of the nearest code in the codebook, and is transmitted to the decoder: $\text{quantize}(E(\mathbf{x})) = \mathbf{e}_k$, where $k = \arg\min_j ||E(\mathbf{x}) - \mathbf{e}_j||$. To learn these mappings, the gradient of the reconstruction error is then back-propagated through the decoder, and to the encoder using the straight-through gradient estimator (Bengio et al., 2013). Besides reconstruction loss, VQ-VAE has two additional terms to align the token space of the codebook with the output of the encoder. The *codebook loss*, which only applies to the codebook variables, brings the selected code $\mathbf{e}$ close to the output of the encoder, $E(\mathbf{x})$. The *commitment loss*, which only applies to the encoder weights, encourages the output of the encoder to stay close to the chosen code to prevent it from fluctuating too frequently from one code vector to another. The overall objective is:

$$\mathcal{L}(\mathbf{x}, D(\mathbf{e})) = ||\mathbf{x} - D(\mathbf{e})||_2^2 + ||\text{sg}[E(\mathbf{x})] - \mathbf{e}||_2^2 + \eta||\text{sg}[\mathbf{e}] - E(\mathbf{x})||_2^2, \quad (1)$$

where $E$ is the encoder function and $D$ is the decoder function. The operator sg refers to a stop-

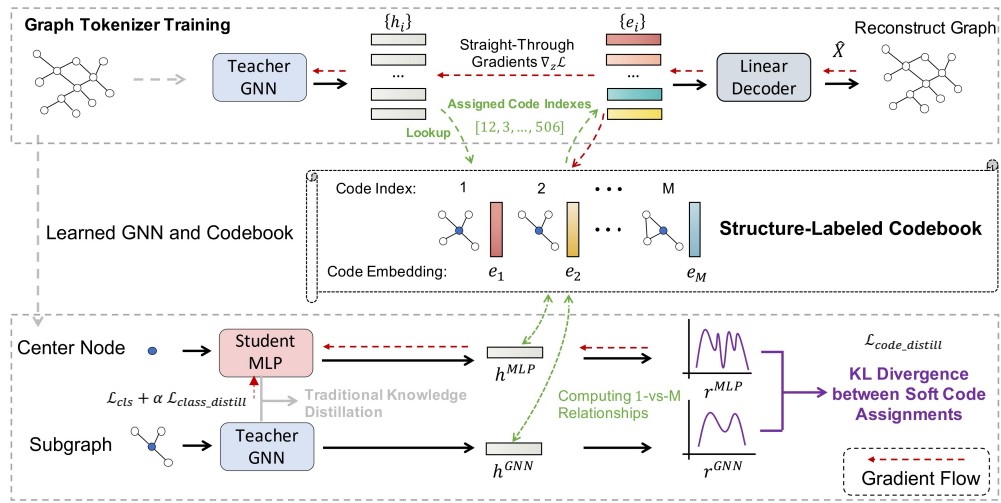

Figure 2: The schematic diagram of VQGRAPH, including graph tokenizer training (**Top**) and structure-aware code-based GNN-to-MLP Distillation (**Bottom**).

gradient operation that blocks gradients from flowing into its argument, and $\eta$ is a hyperparameter which controls the reluctance to change the code corresponding to the encoder output. In this paper, **we explore the potential of VQ-VAE for representing discrete graph-structured data**.

# 4 VQGRAPH

The critical insights of VQGRAPH is learning an expressive graph representation space that directly labels nodes' diverse local neighborhood structures with different code indices for facilitating effective structure-aware GNN-to-MLP distillation. First, we learn a structure-aware graph tokenizer to encode the nodes with diverse local structures to corresponding discrete codes, and constitute a *codebook* (Sec. 4.1). Then we utilize the learned codebook for GNN-to-MLP distillation, and propose a tailored structure-aware distillation objective based on *soft code assignmnets* (Sec. 4.2).

## 4.1 GRAPH TOKENIZER TRAINING

**Labeling Nodes' Local Structure with Discrete Codes** Similar to the tokenization in NLP (Sennrich et al., 2016; Wu et al., 2016), we tokenize the nodes with different neighborhood structures as discrete codes using a variant of VQ-VAE (Van Den Oord et al., 2017), *i.e.*, a graph tokenizer that consists of a GNN encoder and a codebook. More concretely, the nodes $\mathcal{V} = \{\boldsymbol{v}_1, \boldsymbol{v}_2, \cdots, \boldsymbol{v}_n\}$ of a graph $\mathcal{G}$ are tokenized to $\mathcal{Z} = \{z_1, z_2, \cdots, z_n\}$, where the codebook contains $M$ discrete codes. Firstly, the teacher GNN encoder encodes the nodes to nodes embeddings. Next, our graph tokenizer looks up the nearest neighbor code embedding in the codebook for each node embedding $\boldsymbol{h}_i$. Let $\boldsymbol{E} = [\boldsymbol{e}_1, \boldsymbol{e}_2, \cdots, \boldsymbol{e}_M] \in \mathbb{R}^{M \times D}$ denote the codebook embeddings, which are randomly initialized and are then optimized in pretraining. The assigned code of i-th node is:

$$z_i = \text{argmin}_j \|\boldsymbol{h}_i - \boldsymbol{e}_j\|_2, \tag{2}$$

We feed the corresponding codebook embeddings $\{\boldsymbol{e}_{z_1}, \boldsymbol{e}_{z_2}, \cdots, \boldsymbol{e}_{z_n}\}$ to the linear decoder ($p_\psi : \boldsymbol{e}_{z_i} \to \hat{\boldsymbol{v}}_i$) to reconstruct the input graph including both nodes attributes $\boldsymbol{X} \in \mathbb{R}^{N \times D}$ and adjacency matrix $\boldsymbol{A} \in \mathbb{R}^{N \times N}$ for an end-to-end optimization of our graph tokenizer.

**Graph Tokenizer Optimization** We adapt VQ-VAE (first term in Equation (1)) to fit our graph tokenization. In addition, the categorical information is also critical for node representations, thus we integrate it into the optimization of our graph tokenizer:

$$\mathcal{L}_{Rec} = \underbrace{\frac{1}{N}\sum_{i=1}^{N}\left(1 - \frac{\boldsymbol{v}_i^T \hat{\boldsymbol{v}}_i}{\|\boldsymbol{v}_i\| \cdot \|\hat{\boldsymbol{v}}_i\|}\right)^\gamma}_{\text{node reconstruction}} + \underbrace{\left\|\boldsymbol{A} - \sigma(\hat{\boldsymbol{X}} \cdot \hat{\boldsymbol{X}}^T)\right\|_2^2}_{\text{edge reconstruction}}, \tag{3}$$

$$\mathcal{L}_{Tokenizer} = \mathcal{L}_{Rec} + \mathcal{L}_{CE}(\boldsymbol{y}_{\{\boldsymbol{v}_i\}}, \hat{\boldsymbol{y}}_{\{\boldsymbol{e}_{z_i}\}}) + \frac{1}{N}\sum_{i=1}^{N}\|\text{sg}[\boldsymbol{h}_i] - \boldsymbol{e}_{z_i}\|_2^2 + \frac{\eta}{N}\sum_{i=1}^{N}\|\text{sg}[\boldsymbol{e}_{z_i}] - \boldsymbol{h}_i\|_2^2,$$

where $\hat{v} \in \mathbb{R}^D$ and $\hat{X} \in \mathbb{R}^{N \times D}$ denote the predicted node embedding and node embedding matrix, respectively. $\text{sg}[\cdot]$ stands for the stopgradient operator, and the flow of gradients is illustrated in Figure 2. $\mathcal{L}_{Rec}$ denotes the graph reconstruction loss, aiming to preserve node attributes by the first node reconstruction term with the scaled cosine error ($\gamma \geq 1$), and recover graph structures by the second topology reconstruction term. $\mathcal{L}_{CE}$ is the cross-entropy loss between labels $y_{\{v_i\}}$ and the GNN predictions $\hat{y}_{\{e_{z_i}\}}$ that are based on the assigned codes $\{e_{z_i}\}$. In $\mathcal{L}_{Tokenizer}$, the third term is a VQ loss aiming to update the codebook embeddings and the forth term is a commitment loss that encourages the output of the GNN encoder to stay close to the chosen code embedding. $\eta$ is a hyper-parameter set to 0.25 in our experiments. With the learned graph tokenizer, we acquire a powerful codebook that not only directly identifies local graph structures, but also preserves class information for node representations, facilitating later GNN-to-MLP distillation.

**Clarifying Superiority over VQ-VAE and Graph AutoEncoders**  In contrast to vanilla VQ-VAE, we provide a new variant of VQ-VAE for modeling discrete graph data instead of continuous data, and utilize the learned codebook for distillation task instead of generation tasks. In contrast to traditional graph autoencoders (Kipf & Welling, 2016), our model does not suffer from large variance (Van Den Oord et al., 2017). And with our expressive latent codes, we can effectively avoid "posterior collapse" issue which has been problematic with many graph AE models that have a powerful decoder, often caused by latents being ignored. We provide experimental comparison to demonstrate our superiority in Sec. 5.3.

**Scaling to Large-Scale Graphs**  We have introduced the main pipeline of graph tokenizer training based on the entire graph input. Nevertheless, for large-scale industrial applications, one can not feed the whole graph due to the latency constraint (Fey et al., 2021; Bojchevski et al., 2020; Ying et al., 2018; Chen et al., 2020a). Numerous researches choose subgraph-wise sampling methods as a promising class of mini-batch training techniques (Chen et al., 2018b;a; Zeng et al., 2020; Huang et al., 2018; Chiang et al., 2019; Zou et al., 2019; Shi et al., 2023), implicitly covering the global context of graph structure through a number of stochastic re-sampling. We follow this technique to perform large-scale graph tokenizer training. For example, we adopt GraphSAGE (Hamilton et al., 2017) as teacher GNN, we samples the target nodes as a mini-batch $\mathcal{V}_{sample}$ and samples a fixed size set of neighbors for feature aggregation. We utilize the connections between the target nodes as the topology reconstruction target (the second term of $\mathcal{L}_{Rec}$ in Equation (3)) for approximately learning global graph structural information.

### 4.2 STRUCTURE-AWARE CODE-BASED GNN-TO-MLP DISTILLATION

After the graph tokenizer optimization, we obtain a pre-trained teacher GNN encoder and a set of codebook embeddings $E$. We hope to distill the structure knowledge node-by-node from the GNN to a student MLP based on the codebook. Next, we will introduce our tailored structure-aware GNN-to-MLP distillation with the *soft code assignments* over the learned codebook.

**Aligning Soft Code Assignments Between GNN and MLP**  Different from previous class-based distillation methods (Zhang et al., 2022b; Tian et al., 2023b) that constrain on class predictions between GNN and MLP, we utilize a more expressive representation space of graph data, *i.e.*, our structure-aware codebook, and propose code-based distillation to leverage more essential information of graph structures for bridging GNN and MLP. Formally, for each node $v_i$, we have its GNN representation $h_i^{\text{GNN}} \in \mathbb{R}^D$ and the MLP representation $h_i^{\text{MLP}} \in \mathbb{R}^D$. Then we respectively compare their node representations with all $M$ codes of the codebook embeddings $E \in \mathbb{R}^{M \times D}$ and obtain corresponding *soft code assignments* $r_i^{\text{GNN}} \in \mathbb{R}^M$ and $r_i^{\text{MLP}} \in \mathbb{R}^M$:

$$r_i^{\text{GNN}} = \text{COMP}(h_i^{\text{GNN}}, E), \quad r_i^{\text{MLP}} = \text{COMP}(h_i^{\text{MLP}}, E), \quad (4)$$

where $\text{COMP} : [\mathbb{R}^D, \mathbb{R}^{M \times D}] \to \mathbb{R}^M$ can be arbitrary relation module for computing 1-*vs*-$M$ code-wise relations, and such relations can be viewed as assignment possibilities. We use $L_2$ distance in our experiments (more studies in Appendix C.1). Kindly note that the codebook size $M$ can be large, especially for large-scale graphs, thus the soft code assignment of each node contains abundant 1-*vs*-$M$ global structure-discriminative information. Therefore we choose the soft code assignment as

the target for final distillation:

$$\mathcal{L}_{code\_distill} = \frac{1}{N}\sum_{i=1}^{N}\tau^2\,\mathrm{KL}(\boldsymbol{p}_i^{\mathrm{GNN}}\,\|\,\boldsymbol{p}_i^{\mathrm{MLP}}) = \frac{1}{N}\sum_{i=1}^{N}\tau^2\,\boldsymbol{p}_i^{\mathrm{GNN}}\log\frac{\boldsymbol{p}_i^{\mathrm{GNN}}}{\boldsymbol{p}_i^{\mathrm{MLP}}},\qquad(5)$$

where KL refers to Kullback–Leibler divergence with:

$$\boldsymbol{p}_i^{\mathrm{GNN}} = \mathrm{Softmax}(\boldsymbol{r}_i^{\mathrm{GNN}}/\tau),\quad \boldsymbol{p}_i^{\mathrm{MLP}} = \mathrm{Softmax}(\boldsymbol{r}_i^{\mathrm{MLP}}/\tau),\qquad(6)$$

being the scaled code assignments, and $\tau$ is the temperature factor to control the softness. Kindly note that the code assignment is only used for optimizing the training, and we remove it for deploying the pure MLP model. In this way, VQGRAPH is able to effectively distill both local neighborhood structural knowledge and global structure-discriminative ability from GNNs to MLPs, without increasing inference time. The overall training loss of VQGRAPH is composed of classification loss $\mathcal{L}_{cls}$, traditional class-based distillation loss $\mathcal{L}_{class\_distill}$, and our code-based distillation loss, i.e.,

$$\mathcal{L}_{\mathrm{VQGRAPH}} = \mathcal{L}_{cls} + \alpha\mathcal{L}_{class\_distill} + \beta\mathcal{L}_{code\_distill}.\qquad(7)$$

where $\alpha$ and $\beta$ are factors for balancing the losses.

## 5 EXPERIMENTS

**Datasets and Evaluation**  We use five widely used public benchmark datasets (Zhang et al., 2022b; Yang et al., 2021a) (`Citeseer`, `Pubmed`, `Cora`, `A-computer`, and `A-photo`), and two large OGB datasets (Hu et al., 2020a) (`Arxiv` and `Products`) to evaluate the proposed model. In our experiments, we report the mean and standard deviation of ten distinct runs with randomized seeds to ensure robustness and reliability of our findings. **We also extend our VQGRAPH to heterophilic graphs and make performance improvement in Appendix A.2.** We utilize accuracy to gauge model performance. More details are in Appendix B.1.

**Model Architectures**  For fair comparison, we adopt GraphSAGE with GCN aggregation as our teacher model (also as graph tokenizer) and use the same student MLP models for all evaluations following SOTA GLNN (Zhang et al., 2022b) and NOSMOG (Tian et al., 2023b). The codebook size increases accordingly with dataset size (studied in Sec. 5.3). For example, we set 2048 and 8192 for `Cora` and `A-photo`, respectively. More model hyperparameters are detailed in Appendix B.2. We investigate the influence of alternative teacher models, including GCN (Kipf & Welling, 2017), GAT (Veličković et al., 2018), and APPNP (Klicpera et al., 2019), detailed in Appendix C.2.

**Transductive vs. Inductive**  We experiment in two separate settings, transductive (*tran*) and inductive (*ind*), for comprehensive evaluation. In both settings, we first pre-train our graph tokenizer to learn the codebook embeddings $\boldsymbol{E}$ for code-based distillation. For the *tran* setting, we train our models on the labeled graph $\mathcal{G}$, along with the corresponding feature matrix $\boldsymbol{X}^L$ and label vector $\boldsymbol{Y}^L$, before evaluating their performance on the unlabeled data $\boldsymbol{X}^U$ and $\boldsymbol{Y}^U$. Soft labels, soft code assignments are generated for all nodes within the graph (i.e., $\boldsymbol{y}_v^{soft}$, $\boldsymbol{r}_v^{GNN}$, $\boldsymbol{r}_v^{MLP}$ for $v \in \mathcal{V}$). As for *ind*, we follow the methodology of prior work (Tian et al., 2023b) in randomly selecting 20% of the data for inductive evaluation. Specifically, we divide the unlabeled nodes $\mathcal{V}^U$ into two separate yet non-overlapping subsets, observed and inductive (i.e., $\mathcal{V}^U = \mathcal{V}_{obs}^U \sqcup \mathcal{V}_{ind}^U$), producing three distinct graphs, $\mathcal{G} = \mathcal{G}^L \sqcup \mathcal{G}_{obs}^U \sqcup \mathcal{G}_{ind}^U$, wherein there are no shared nodes. In training, the edges between $\mathcal{G}^L \sqcup \mathcal{G}_{obs}^U$ and $\mathcal{G}_{ind}^U$ are removed, while they are leveraged during inference to transfer positional features via average operator (Hamilton et al., 2017). Node features and labels are partitioned into three disjoint sets, *i.e.*, $\boldsymbol{X} = \boldsymbol{X}^L \sqcup \boldsymbol{X}_{obs}^U \sqcup \boldsymbol{X}_{ind}^U$ and $\boldsymbol{Y} = \boldsymbol{Y}^L \sqcup \boldsymbol{Y}_{obs}^U \sqcup \boldsymbol{Y}_{ind}^U$. Soft labels and soft code assignments are generated for nodes within the labeled and observed subsets (i.e., $\boldsymbol{y}_v^{soft}$, $\boldsymbol{r}_v^{GNN}$, $\boldsymbol{r}_v^{MLP}$ for $v \in \mathcal{V}^L \sqcup \mathcal{V}_{obs}^U$). We provide code and models in the supplementary material.

### 5.1 MAIN RESULTS

**GNN-to-MLP Distillation**  We compare VQGRAPH to other state-of-the-art GNN-to-MLP distillation methods GLNN and NOSMOG with same experimental settings, and use distilled MLP models for evaluations. We first consider the standard transductive setting, enabling direct comparison with previously published literature (Zhang et al., 2022b; Hu et al., 2020b; Yang et al., 2021a). As depicted in Tab. 1, VQGRAPH outperforms all baselines including teacher GNN models across

Table 1: Node classification results under the standard setting, results show accuracy (higher is better). $\Delta_{GNN}$, $\Delta_{MLP}$, $\Delta_{NOSMOG}$ represents the difference between VQGRAPH and GNN, MLP, NOSMOG, respectively. GLNN and NOSMOG are the SOTA GNN-to-MLP distillation methods.

| Datasets | SAGE | MLP | GLNN | NOSMOG | VQGRAPH | $\Delta_{GNN}$ | $\Delta_{MLP}$ | $\Delta_{NOSMOG}$ |
|---|---|---|---|---|---|---|---|---|
| Citeseer | $70.49 \pm 1.53$ | $58.50 \pm 1.86$ | $71.22 \pm 1.50$ | $73.78 \pm 1.54$ | $\mathbf{76.08 \pm 0.55}$ | ↑7.93% | ↑30.05% | ↑3.18% |
| Pubmed | $75.56 \pm 2.06$ | $68.39 \pm 3.09$ | $75.59 \pm 2.46$ | $77.34 \pm 2.36$ | $\mathbf{78.40 \pm 1.71}$ | ↑3.76% | ↑14.64% | ↑1.37% |
| Cora | $80.64 \pm 1.57$ | $59.18 \pm 1.60$ | $80.26 \pm 1.66$ | $83.04 \pm 1.26$ | $\mathbf{83.93 \pm 0.87}$ | ↑4.08% | ↑41.82% | ↑1.07% |
| A-computer | $82.82 \pm 1.37$ | $67.62 \pm 2.21$ | $82.71 \pm 1.18$ | $84.04 \pm 1.01$ | $\mathbf{85.17 \pm 1.29}$ | ↑2.84% | ↑25.95% | ↑1.34% |
| A-photo | $90.85 \pm 0.87$ | $77.29 \pm 1.79$ | $91.95 \pm 1.04$ | $93.36 \pm 0.69$ | $\mathbf{94.21 \pm 0.45}$ | ↑3.70% | ↑21.89% | ↑0.91% |
| Arxiv | $70.73 \pm 0.35$ | $55.67 \pm 0.24$ | $63.75 \pm 0.48$ | $71.65 \pm 0.29$ | $\mathbf{72.43 \pm 0.20}$ | ↑2.40% | ↑30.11% | ↑0.93% |
| Products | $77.17 \pm 0.32$ | $60.02 \pm 0.10$ | $63.71 \pm 0.31$ | $78.45 \pm 0.38$ | $\mathbf{79.17 \pm 0.21}$ | ↑2.59% | ↑31.91% | ↑0.92% |

Table 2: Node classification results in a production scenario with both **inductive** and **transductive** settings. *ind* indicates the results on $\mathcal{V}_{ind}^{U}$, *tran* indicates the results on $\mathcal{V}_{tran}^{U}$, and *prod* indicates the interpolated production results of both *ind* and *tran*.

| Datasets | Eval | SAGE | MLP | GLNN | NOSMOG | VQGRAPH | $\Delta_{GNN}$ | $\Delta_{MLP}$ | $\Delta_{NOSMOG}$ |
|---|---|---|---|---|---|---|---|---|---|
| Citeseer | *prod* | 68.06 | 58.49 | 69.08 | 70.60 | **73.76** | ↑8.37% | ↑26.11% | ↑5.80% |
|  | *ind* | $69.14 \pm 2.99$ | $59.31 \pm 4.56$ | $68.48 \pm 2.38$ | $70.30 \pm 2.30$ | $\mathbf{72.93 \pm 1.78}$ | ↑5.48% | ↑22.96% | ↑3.74% |
|  | *tran* | $67.79 \pm 2.80$ | $58.29 \pm 1.94$ | $69.23 \pm 2.39$ | $70.67 \pm 2.25$ | $\mathbf{74.59 \pm 1.94}$ | ↑10.03% | ↑27.96% | ↑7.74% |
| Pubmed | *prod* | 74.77 | 68.39 | 74.67 | 75.82 | **76.92** | ↑2.86% | ↑12.47% | ↑1.45% |
|  | *ind* | $75.07 \pm 2.89$ | $68.28 \pm 3.25$ | $74.52 \pm 2.95$ | $75.87 \pm 3.32$ | $\mathbf{76.71 \pm 2.76}$ | ↑2.18% | ↑12.35% | ↑1.11% |
|  | *tran* | $74.70 \pm 2.33$ | $68.42 \pm 3.06$ | $74.70 \pm 2.75$ | $75.80 \pm 3.06$ | $\mathbf{77.13 \pm 3.01}$ | ↑3.25% | ↑12.73% | ↑1.75% |
| Cora | *prod* | 79.53 | 59.18 | 77.82 | 81.02 | **81.68** | ↑2.70% | ↑38.02% | ↑0.81% |
|  | *ind* | $81.03 \pm 1.71$ | $59.44 \pm 3.36$ | $73.21 \pm 1.50$ | $81.36 \pm 1.53$ | $\mathbf{82.20 \pm 1.32}$ | ↑1.44% | ↑38.29% | ↑1.03% |
|  | *tran* | $79.16 \pm 1.60$ | $59.12 \pm 1.49$ | $78.97 \pm 1.56$ | $80.93 \pm 1.65$ | $\mathbf{81.15 \pm 1.25}$ | ↑2.51% | ↑37.26% | ↑0.27% |
| A-computer | *prod* | 82.73 | 67.62 | 82.10 | 83.85 | **84.16** | ↑1.73% | ↑24.46% | ↑0.37% |
|  | *ind* | $82.83 \pm 1.51$ | $67.69 \pm 2.20$ | $80.27 \pm 2.11$ | $84.36 \pm 1.57$ | $\mathbf{85.73 \pm 2.04}$ | ↑3.50% | ↑26.65% | ↑1.62% |
|  | *tran* | $82.70 \pm 1.34$ | $67.60 \pm 2.23$ | $82.56 \pm 1.80$ | $83.72 \pm 1.44$ | $\mathbf{84.56 \pm 1.81}$ | ↑2.25% | ↑25.08% | ↑1.00% |
| A-photo | *prod* | 90.45 | 77.29 | 91.34 | 92.47 | **93.05** | ↑2.87% | ↑20.39% | ↑0.62% |
|  | *ind* | $90.56 \pm 1.47$ | $77.44 \pm 1.50$ | $89.50 \pm 1.12$ | $92.61 \pm 1.09$ | $\mathbf{93.11 \pm 0.89}$ | ↑2.82% | ↑20.24% | ↑0.54% |
|  | *tran* | $90.42 \pm 0.68$ | $77.25 \pm 1.90$ | $91.80 \pm 0.49$ | $92.44 \pm 0.51$ | $\mathbf{92.96 \pm 1.02}$ | ↑2.81% | ↑20.34% | ↑0.56% |
| Arxiv | *prod* | 70.69 | 55.35 | 63.50 | 70.90 | **71.43** | ↑1.05% | ↑29.05% | ↑0.75% |
|  | *ind* | $70.69 \pm 0.58$ | $55.29 \pm 0.63$ | $59.04 \pm 0.46$ | $70.09 \pm 0.55$ | $\mathbf{70.86 \pm 0.42}$ | ↑0.24% | ↑28.16% | ↑1.10% |
|  | *tran* | $70.69 \pm 0.39$ | $55.36 \pm 0.34$ | $64.61 \pm 0.15$ | $71.10 \pm 0.34$ | $\mathbf{72.03 \pm 0.56}$ | ↑1.90% | ↑30.11% | ↑1.31% |
| Products | *prod* | 76.93 | 60.02 | 63.47 | 77.33 | **77.93** | ↑1.30% | ↑29.84% | ↑0.77% |
|  | *ind* | $77.23 \pm 0.24$ | $60.02 \pm 0.09$ | $63.38 \pm 0.33$ | $77.02 \pm 0.19$ | $\mathbf{77.50 \pm 0.25}$ | ↑0.35% | ↑29.12% | ↑0.62% |
|  | *tran* | $76.86 \pm 0.27$ | $60.02 \pm 0.11$ | $63.49 \pm 0.31$ | $77.41 \pm 0.21$ | $\mathbf{78.36 \pm 0.13}$ | ↑1.95% | ↑30.56% | ↑1.23% |

all datasets. Specifically, VQGRAPH improves performance **by an average of 3.90% compared to its teacher GNN**, highlighting its ability to capture superior structural information without relying on explicit graph structure input. Comparing VQGRAPH to NOSMOG, our proposed model **achieves an average improvement of 1.39%** across both small- and large-scale graph datasets. Further model analysis of VQGRAPH is presented in Sec. 5.3.

**Experiments in Inductive and Transductive Settings** To gain deeper insights into the effectiveness of VQGRAPH, we conduct experiments in a realistic production (*prod*) scenario that involves both inductive (*ind*) and transductive (*tran*) settings across multiple datasets, as detailed in Tab. 2. Our experimental results demonstrate that VQGRAPH consistently achieves superior performance compared to the teacher model and baseline methods across all datasets and settings. Specifically, our proposed method outperforms GNN across all datasets and settings with **an average improvement of 2.93%**, demonstrating its superior efficacy of our learned code-based representation space in capturing graph structural information, even on large-scale datasets. Furthermore, when compared to MLP and NOSMOG, VQGRAPH consistently achieves significant performance improvements, **with average gains of 25.81% and 1.6%**, respectively, across all datasets and settings.

## 5.2 MODEL ANALYSIS

**Trade-off between Performance and Inference Time** To demonstrate its efficiency and capacity of our VQGRAPH, we visualize the trade-off between node classification accuracy and model inference time on Citeseer dataset in Figure 3. Our results indicate that achieves a highest accuracy of 76% while maintaining a fast inference time of 1.45ms. Compared to the other models with similar inference time, VQGRAPH significantly **outperforms NOSMOG and MLPs by 3.12% and 30.05% in average accuracy**, respectively. For those models having comparable performance with VQGRAPH, they require a considerable amount of inference time, e.g., 2 layers GraphSAGE (SAGE-L2) needs 152.31ms and 3 layers GraphSAGE (SAGE-L3) needs 1201.28ms, making them

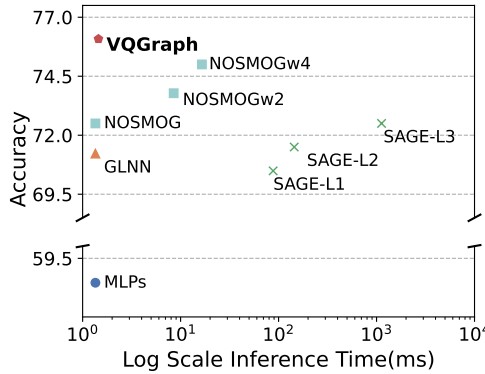

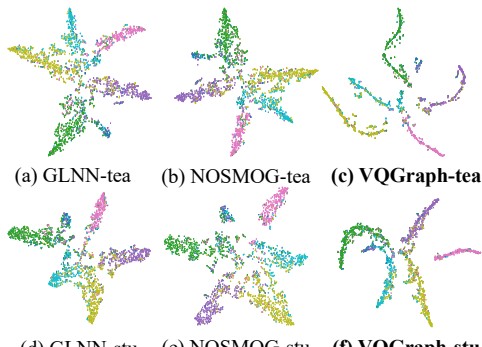

Figure 3: Accuracy vs. Inference Time.

Figure 4: t-SNE visualization of learned node representations, colors denotes different classes.

unsuitable for real-world applications. This **makes VQGRAPH 105× faster than SAGE-L2 and 828× faster than SAGE-L3**. Although increasing the hidden size of NOSMOG slightly improves its performance, NOSMOGw2 (2-times wider than NOSMOG) and NOSMOGw4 still perform worse than VQGRAPH with more inference time, demonstrating the superior efficiency of our VQGRAPH.

**Compactness of Learned Node Representation Space** We use t-SNE (Van der Maaten & Hinton, 2008) to visualize the node representation spaces of both teacher GNN and distilled student MLP models with different methods, and put the results in Figure 4. Our VQGRAPH provides a better teacher GNN model than GLNN and NOSMOG, and the node representations of same classes have a more compact distribution. The representations extracted by our distilled MLP model also have a more compact distribution. We attribute these to our expressive code-based representation space, providing more structure-aware representations for classifying nodes. Besides, our new code-based distillation strategy can effectively deliver both graph structural information and categorial information from GNN to MLP, guaranteeing the compactness of MLP's representation space.

**Consistency between Model Predictions and Global Graph Topology** Here we corroborate the superiority of VQGRAPH over GNNs, MLPs, GLNN and NOSMOG in capturing global graph structural information, which is complementary to the above analysis on local structure awareness. We use the cut value to effectively evaluate the alignment between model predictions and graph topology as (Zhang et al., 2022b; Tian et al., 2023b) based on the approximation for the min-cut problem (Bianchi

Table 3: The cut value. VQGRAPH predictions are more consistent with the graph topology than GNN, MLP, GLNN and the state-of-the-art method NOSMOG.

| Datasets | SAGE | MLP | GLNN | NOSMOG | **VQGRAPH** |
|---|---|---|---|---|---|
| Citeseer | 0.9535 | 0.8107 | 0.9447 | 0.9659 | **0.9786** |
| Pubmed | 0.9597 | 0.9062 | 0.9298 | 0.9641 | **0.9883** |
| Cora | 0.9385 | 0.7203 | 0.8908 | 0.9480 | **0.9684** |
| A-computer | 0.8951 | 0.6764 | 0.8579 | 0.9047 | **0.9190** |
| A-photo | 0.9014 | 0.7099 | 0.9063 | 0.9084 | **0.9177** |
| Arxiv | 0.9052 | 0.7252 | 0.8126 | 0.9066 | **0.9162** |
| Products | 0.9400 | 0.7518 | 0.7657 | 0.9456 | **0.9571** |
| Average | 0.9276 | 0.7572 | 0.8725 | 0.9348 | **0.9493** |

et al., 2019). The min-cut problem divides nodes $\mathcal{V}$ into $K$ disjoint subsets by removing the minimum number of edges. Correspondingly, the min-cut problem can be expressed as: $\max \frac{1}{K} \sum_{k=1}^{K} (\boldsymbol{C}_k^T \boldsymbol{A} \boldsymbol{C}_k)/(\boldsymbol{C}_k^T \boldsymbol{D} \boldsymbol{C}_k)$, where $\boldsymbol{C}$ is the node class assignment, $\boldsymbol{A}$ is the adjacency matrix, and $\boldsymbol{D}$ is the degree matrix. Therefore, cut value is defined as follows: $\mathcal{CV} = tr(\hat{\boldsymbol{Y}}^T \boldsymbol{A} \hat{\boldsymbol{Y}})/tr(\hat{\boldsymbol{Y}}^T \boldsymbol{D} \hat{\boldsymbol{Y}})$, where $\hat{\boldsymbol{Y}}$ is the model prediction output, and the cut value $\mathcal{CV}$ indicates the consistency between the model predictions and the graph topology. We report the cut values for various models in the transductive setting in Tab. 3. The average $\mathcal{CV}$ achieved by VQGRAPH is **0.9493**, while SAGE, GLNN, and NOSMOG have average $\mathcal{CV}$ values of 0.9276, 0.8725, and 0.9348, respectively. We find VQGRAPH obtains the highest cut value, indicating the superior global structure-capturing ability over GNN and SOTA GNN-to-MLP distillation methods.

## 5.3 ABLATION STUDY

**Influence of the Codebook Size** We analyze the influence of the codebook size of our graph tokenizer. From Figure 5(i), we observe that changing codebook size can significantly influence the performance for our distilled MLP model. Too small a value can not have enough expressiveness

for preserving graph structures while too large a value will lead to code redundancy impairing the performance. Morever, VQGRAPH has different optimal codebook sizes for various datasets as in Figure 5(ii). Another interesting observation is the graph with more nodes or edges tends to require a larger codebook size to achieve optimal distillation results. In our VQGraph, the size of the codebook is mainly determined by the complexity of the graph data, considering both nodes and edges which produce different local substructures. Thus, our codebook size is still small compared to the exponential graph topological complexity, demonstrating the expressiveness of our codebook. Taking `Cora` as an example, our codebook size is 2048, but it contains 2485 nodes with an average degree about 4 which can theoretically result in $O(2485^4)$ possible 1-hop substructure patterns.

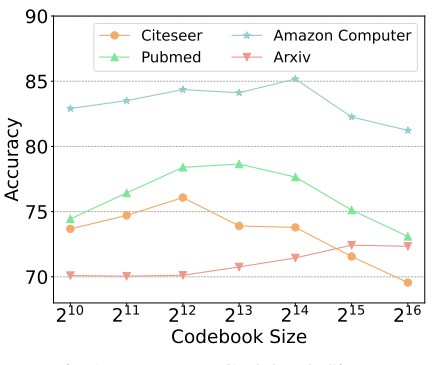
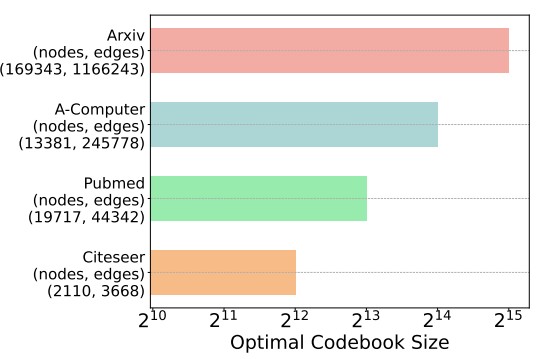

(i) Accuracy vs. Codebook Size.  (ii) Optimal codebook size for various datasets.

Figure 5: Influence of the codebook size.

**Contribution Analysis of VQGRAPH**  We design experiments to make statistical analysis on the contributions of our VQGRAPH. The results are presented in Tab. 4, and we observe that compared to traditional class-based distillation, both *Only-VQ* and VQGRAPH promote the average accuracy, suggesting that both graph tokenizer and soft code assignments have vital impacts on final performance. Moreover, comparing *AE+class-based* to *class-based*, we find adding structure awareness slightly improves GNN-to-MLP distillation. Our designed graph VQ-VAE efficiently improves the GNN-to-MLP distillation results more significantly than classic graph (Variational)AE, because we directly learn numerous structure-aware codes to enrich the expressiveness of node representations. Our *VQ+code-based* distillation (denoted as VQGRAPH) substantially improves node classification performance over *Only-VQ* across all datasets, demonstrating the superiority of our new structure-aware distillation targets over soft labels. Please refer to Appendix C for more ablation studies.

Table 4: *Class-based* denotes only using soft labels for distillation (e.g., GLNN), *AE+Class-based* denotes adding classic graph Auto-Encoder (Kipf & Welling, 2016) for structure awareness. *Only-VQ* denotes using VQ for training teacher but using soft labels for distillation. $\Delta_{\text{Only-VQ}}$, $\Delta_{\text{VQGRAPH}}$ represents the differences between Only-VQ, VQGRAPH and *Class-based*.

| Datasets | GNN | Class-based | AE+Class-based | **Only-VQ (ours)** | **VQGRAPH (ours)** | $\Delta_{\text{Only-VQ}}$ | $\Delta_{\text{VQGRAPH}}$ |
|---|---|---|---|---|---|---|---|
| Citeseer | 70.49 ± 1.53 | 71.22 ± 1.54 | 71.65 ± 0.69 | **74.96 ± 1.50** | **76.08 ± 0.55** | ↑ 5.25% | ↑ 6.82% |
| Pubmed | 75.56 ± 2.06 | 75.59 ± 2.46 | 76.56 ± 1.23 | **77.86 ± 2.46** | **78.40 ± 1.71** | ↑ 3.00% | ↑ 3.71% |
| Cora | 80.64 ± 1.57 | 80.26 ± 1.66 | 81.11 ± 1.01 | **82.48 ± 0.46** | **83.93 ± 0.87** | ↑ 2.77% | ↑ 4.57% |
| A-computer | 82.82 ± 1.37 | 82.71 ± 1.18 | 83.01 ± 1.18 | **84.06 ± 1.18** | **85.17 ± 1.29** | ↑ 1.63% | ↑ 2.97% |
| A-photo | 90.85 ± 0.87 | 91.95 ± 1.04 | 92.06 ± 0.69 | **93.86 ± 1.04** | **94.21 ± 0.45** | ↑ 2.08% | ↑ 2.45% |
| Arxiv | 70.73 ± 0.35 | 63.75 ± 0.48 | 70.10 ± 1.02 | **70.75 ± 0.48** | **72.43 ± 0.20** | ↑ 10.98% | ↑ 13.61% |
| Products | 77.17 ± 0.32 | 67.71 ± 0.31 | 77.65 ± 0.98 | **78.71 ± 0.31** | **79.17 ± 0.21** | ↑ 16.25% | ↑ 16.93% |

## 6  CONCLUSION

In this paper, we improve the expressiveness of existing graph representation space by directly labeling nodes' diverse local structures with a codebook, and utilizing the codebook for facilitating structure-aware GNN-to-MLP distillation. Extensive experiments on seven datasets demonstrate that our VQGRAPH can significantly improve GNNs by **3.90%**, MLPs by **28.05%**, and the state-of-the-art GNN-to-MLP distillation method by **1.39%** on average accuracy, while maintaining a fast inference speed of **828×** compared to GNNs. Furthermore, we present additional visualization and statistical analyses as well as ablation studies to demonstrate the superiority of the proposed model.

ACKNOWLEDGEMENT

This work was supported by the National Natural Science Foundation of China (No.U22B2037 and U23B2048).

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

## A   MORE ANALYSIS AND RESULTS

### A.1   CONNECTION BETWEEN LOCAL GRAPH STRUCTURE AND LEARNED CODEBOOK

To better illustrate the connection between local graph structure and the codebook learned by our graph VQ-VAE, we conduct node-centered subgraph retrieval in the learned MLP representation spaces of NOSMOG and our VQGRAPH. Specifically, we extract the representation with distilled MLP model for a query node in `Citeseer`. We demonstrate 4 subgraphs centered at the nodes that are most similar to the query node representation with the cosine similarities in the whole graph in Figure 6. As can be observed, the identical token IDs yield similar substructures, indicating the representation similarities of VQGRAPH are approximately aligned with the local subgraph similarities, which is denoted as graph edit distance (GED) Sanfeliu & Fu (1983); Bunke & Allermann (1983); Gao et al. (2010) and is one of the most popular graph matching methods. In contrast, the representations extracted from NOSMOG fail to sufficiently model and reflect the structural similarities between subgraphs. Hence, VQGRAPH is necessary and effective for addressing structure-complex graph tasks. Moreover, we observe that VQGRAPH can be aware of the minor neighborhood structural difference despite the nodes with the same class, demonstrating the more fine-grained expressiveness of our token-based distillation, which further helps with more accurate classification.

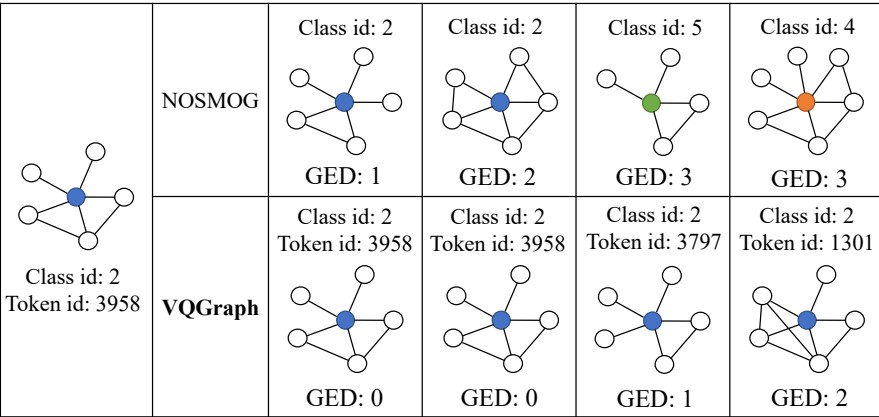

Figure 6: The query node and 4 closest nodes in distilled MLP representation space with corresponding subgraphs.

### A.2   VQGRAPH FOR HETEROPHILIC GRAPHS

In addition to homophilic graphs, we extend our proposed method to heterophilic graphs for further evaluation. We consider 2 heterophilic datasets, `Texas` and `Cornell`, realsed by Pei et al. (2020a). `Texas` and `Cornell` are web page datasets collected from computer science departments of various universities. In these datasets, nodes are web pages and edges represent hyperlinks between them. Bag-of-words representations are taken as nodes' feature vectors. The task is to classify the web pages into five categories including student, project, course, staff and faculty. Basic statistics of the datasets are shown in Tab. 5. The Edge Hom. (Zhu et al. (2020)) is defined as the fraction of edges that connect nodes with the same label.

Table 5: Heterophilic Dataset Statistics.

| Dataset | # Nodes | # Edges | # Features | # Classes | # Edge Hom. |
|---------|---------|---------|-----------|-----------|-------------|
| Texas   | 183     | 295     | 1,703     | 5         | 0.11        |
| Cornell | 183     | 280     | 1,703     | 5         | 0.30        |

We incorporate our structure-aware code embeddings with two state-of-the-art GNN models known for their efficacy on heterophilic graphs, namely ACMGCN (Luan et al. (2022)) and GCNII (Chen

et al. (2020c)), along with MLP. We compare with these baselines in Tab. 6. The results of these experiments demonstrate that our graph VQ-VAE can generalize to different heterophilic GNN/MLP architectures and consistently improve their performances.

Table 6: Results on Heterophilic Graphs.

|  | Texas | Cornell |
|---|---|---|
| ACMGCN | 86.49 | 84.05 |
| ACMGCN with our Graph VQ-VAE | **87.03** | **84.59** |
| MLP | 75.68 | 76.38 |
| MLP with our Graph VQ-VAE | **77.93** | **78.29** |
| GCNII | 76.73 | 76.49 |
| GCNII with our Graph VQ-VAE | **79.04** | **78.29** |

### A.3 ANALYSIS ON CODEBOOK ENTRIES

In the training stage, we tokenize each node with different neighborhood structures as discrete codes using Graph VQ-VAE. Now we take a closer look at codebook entries for different classes on Pubmed for in-depth analysis (Tabs. 7 and 8). There is little difference in the number of codes with each class and there is a uniform distribution of the nodes for each class, demonstrating the learning capacity of our model. Meanwhile, the results demonstrate that there is a small code overlap among different classes, which indicates the code distributions learned for different classes are mutually independent. This finding validates that our graph VQ-VAE enables distinct separation of the nodes with different class labels in the representation space, facilitating a better knowledge transfer in GNN-to-MLP distillation.

Table 7: Codebook Entries for Each Class in `Pubmed`.

|  | Class 1 | Class 2 | Class 3 |
|---|---|---|---|
| Code Entries | 2,560 | 2,623 | 2,591 |

Table 8: The Overlapping Codes (%) for Each Paired Classes of `Pubmed`.

|  | Class 1 | Class 2 | Class 3 |
|---|---|---|---|
| Class 1 | * | 0.9% | 0.8% |
| Class 2 |  | * | 0.6% |
| Class 3 |  |  | * |

## B EXPERIMENT DETAILS

### B.1 DATASETS

The statistics of the seven public benchmark datasets used in our experiments are shown in Tab. 9. For all datasets, we follow the setting in the original paper to split the data. Specifically, for the five small datasets (i.e., Cora, Citeseer, Pubmed, A-computer, and A-photo), we use the splitting strategy in the CPF paper (Yang et al., 2021b), where each random seed corresponding to a different split. For the two OGB large datasets (i.e., Arxiv and Products), we follow the official splits in Hu et al. (2020a) based on time and popularity, respectively. We introduce each dataset as follows:

- `Citeseer` (Sen et al., 2008) is a benchmark citation dataset consisting of scientific publications, with the configuration similar to the Cora dataset. The dictionary contains 3,703 unique words. Citeseer dataset has the largest number of features among all datasets used in this paper.

- `Pubmed` (Namata et al., 2012) is a citation benchmark consisting of diabetes-related articles from the PubMed database. The node features are TF/IDF-weighted word frequency, and the label indicates the type of diabetes covered in this article.

- `Cora` (Sen et al., 2008) is a benchmark citation dataset composed of scientific publications. Each node in the graph represents a publication whose feature vector is a sparse bag-of-words with 0/1-values indicating the absence/presence of the corresponding word from the word dictionary. Edges represent citations between papers, and labels indicate the research field of each paper.

- `A-computer` and `A-photo` (Shchur et al., 2018) are two benchmark datasets that are extracted from Amazon co-purchase graph. Nodes represent products, edges indicate whether two products are frequently purchased together, features represent product reviews encoded by bag-of-words, and labels are predefined product categories.

- `Arxiv` (Hu et al., 2020a) is a benchmark citation dataset composed of Computer Science arXiv papers. Each node is an arXiv paper and each edge indicates that one paper cites another one. The node features are average word embeddings of the title and abstract.

- `Products` (Hu et al., 2020a) is a benchmark Amazon product co-purchasing network dataset. Nodes represent products sold on Amazon, and edges between two products indicate that the products are purchased together. The node features are bag-of-words features from the product descriptions.

Table 9: Dataset Statistics.

| Dataset | # Nodes | # Edges | # Features | # Classes |
|---|---|---|---|---|
| Citeseer | 2,110 | 3,668 | 3,703 | 6 |
| Pubmed | 19,717 | 44,324 | 500 | 3 |
| Cora | 2,485 | 5,069 | 1,433 | 7 |
| A-computer | 13,381 | 245,778 | 767 | 10 |
| A-photo | 7,487 | 119,043 | 745 | 8 |
| Arxiv | 169,343 | 1,166,243 | 128 | 40 |
| Products | 2,449,029 | 61,859,140 | 100 | 47 |

## B.2 MODEL HYPERPARAMETERS AND IMPLEMENTATION DETAILS

The code and trained models are provided in the supplementary material.

**Production Scenario** Regarding to the production scenario, we recognize that real-world deployment of models necessitates the ability to generate predictions for new data points as well as reliably maintain performance on existing ones. Therefore, we apply our method in a realistic production setting to better understand the effectiveness of GNN-to-MLP models, encompassing both transductive and inductive predictions. In a real-life production environment, it is common for a model to be retrained periodically. The holdout nodes in the inductive set represent new nodes that have entered the graph between two training periods. To mitigate potential randomness and to evaluate generalizability more effectively, we employ a test dataset $V_{ind}^U$, which contains 20% of the test data, and another dataset $V_{obs}^U$, containing the remaining 80% of the test data. This setup allows us to evaluate the model's performance on both observed unlabeled nodes (transductive prediction) and newly introduced nodes (inductive prediction), reflecting real-world inference scenarios. We further provide three sets of results: "tran" refers to results on $V_{obs}^U$, "ind" refers to $V_{ind}^U$, and "prod" refers to an interpolated value between tran and ind, according to the split rate, indicating the performance in real production settings.

**Graph Tokenizer Pre-Training** We first train our graph tokenizer which contains a teacher GNN model and a learnable codebook, then use both the GNN and the codebook to conduct GNN-to-MLP distillation. The hyperparameters of GNN models on each dataset are taken from the best hyperparameters provided by the CPF paper (Yang et al., 2021b) (Tab. 10) and the OGB official examples (Hu et al., 2020a) (Tab. 11). We provide the hyperparameters of our codebooks for different datasets

in Tab. 12. Kindly note that the VQ procedure is inserted between the encoder and the classifier of the GNN model, and the dimensions of code embedding and GNN feature are same for convenient assignment. And the code embeddings are initialized with uniform distribution. During the training of our graph tokenizer, VQGRAPH additionally includes two separate linear decoder layers to decode node attributes and graph topology, respectively, based on assigned code embeddings.

**Code-Based GNN-to-MLP Distillation**  We freeze the parameters of both the pre-trained teacher GNN model and the learned codebook embeddings for our code-based GNN-to-MLP distillation. For the student MLP in VQGRAPH, unless otherwise specified with -w$i$ or -L$i$, we set the number of layers and the hidden dimension of each layer to be the same as the teacher GNN, so their total number of parameters stays the same as the teacher GNN (in Tab. 12). In distillation, the teacher GNN not only delivers the soft labels with respect to node classification to the MLP, but also produces its corresponding soft code assignment for the MLP. Kindly recall that we use codebook embeddings to compute code assignments for the MLP only in training process, and we remove this assigning procedure for deploying the distilled MLP in testing process, which does not increase inference time. In practice, categorial and structural information might be of different importance to the distillation in various graph datasets. We correspondingly modify their loss weights for better performance.

Table 10: Hyperparameters for GNNs on five datasets from the CPF paper.

|  | SAGE | GCN | GAT | APPNP |
|---|---|---|---|---|
| # layers | 2 | 2 | 2 | 2 |
| hidden dim | 128 | 64 | 64 | 64 |
| learning rate | 0.01 | 0.01 | 0.01 | 0.01 |
| weight decay | 0.0005 | 0.001 | 0.01 | 0.01 |
| dropout | 0 | 0.8 | 0.6 | 0.5 |
| fan out | 5,5 | - | - | - |
| attention heads | - | - | 8 | - |
| power iterations | - | - | - | 10 |

Table 11: Hyperparameters for GraphSAGE on OGB datasets.

| Dataset | Arxiv | Products |
|---|---|---|
| # layers | 3 | 3 |
| hidden dim | 256 | 256 |
| learning rate | 0.01 | 0.003 |
| weight decay | 0 | 0 |
| dropout | 0.2 | 0.5 |
| normalization | batch | batch |
| fan out | [5, 10, 15] | [5, 10, 15] |

Table 12: Hyperparameters of VQGRAPH.

|  | Citeseer | Pubmed | Cora | A-computer | A-photo | Arxiv | Products |
|---|---|---|---|---|---|---|---|
| tokenizer attribute decoder layers | 1 | 1 | 1 | 1 | 1 | 1 | 1 |
| tokenizer topology decoder layers | 1 | 1 | 1 | 1 | 1 | 1 | 1 |
| codebook size | 4096 | 8192 | 2048 | 16384 | 8192 | 32768 | 32,768 |
| $\tau$ in $\mathcal{L}_{code\_distill}$ | 4 | 4 | 4 | 4 | 4 | 4 | 4 |
| MLP layers | 2 | 2 | 2 | 2 | 2 | 3 | 3 |
| hidden dim | 128 | 128 | 128 | 128 | 128 | 256 | 256 |
| learning rate | 0.01 | 0.01 | 0.005 | 0.003 | 0.001 | 0.01 | 0.003 |
| weight decay | 0.005 | 0.001 | 0.001 | 0.005 | 0.001 | 0 | 0 |
| dropout | 0.6 | 0.1 | 0.4 | 0.1 | 0.1 | 0.2 | 0.5 |
| $\alpha$ for $\mathcal{L}_{class\_distill}$ | 1 | 1 | 1 | 1 | 1 | 1 | 1 |
| $\beta$ for $\mathcal{L}_{code\_distill}$ | 1e-8 | 1e-8 | 1e-8 | 1e-8 | 1e-8 | 1e-8 | 1e-8 |

# C  MORE ABLATION STUDIES

## C.1  VQGRAPH WITH DIFFERENT RELATION MODULES IN EQUATION (4)

In Sec. 4.2, we compute 1-*vs*-$M$ token-wise relations in Equation (4) using $L_2$ distance as the relation module COMP$(\cdot, \cdot)$. Other relation modules can also be applied, and we thus conduct experiments based on cosine similarity with the results depicted in Tab. 13. Experiments across different datasets illustrate that VQGRAPH with different relation modules can consistently improve the code distillation performance of VQGRAPH. The influence of relation module on VQGRAPH's overall performance is very minimal, with $L_2$ distance exhibiting a slight advantage of approximately 0.2% on average accuracy.

Table 13: VQGRAPH with diffrent relation modules.

| Datasets | GNN | w/ Cosine Similarity | w/ $L_2$ **distance** | $\Delta$ |
|---|---|---|---|---|
| Citeseer | $70.49 \pm 1.53$ | $75.80 \pm 0.66$ | $\mathbf{76.08 \pm 0.55}$ | ↑0.12% |
| Pubmed | $75.56 \pm 2.06$ | $78.12 \pm 1.23$ | $\mathbf{78.40 \pm 1.71}$ | ↑0.23% |
| Cora | $80.64 \pm 1.57$ | $83.61 \pm 0.70$ | $\mathbf{83.93 \pm 0.87}$ | ↑0.20% |
| A-computer | $82.82 \pm 1.37$ | $84.96 \pm 1.06$ | $\mathbf{85.17 \pm 1.29}$ | ↑0.20% |
| A-photo | $90.85 \pm 0.87$ | $94.05 \pm 0.29$ | $\mathbf{94.21 \pm 0.45}$ | ↑0.18% |
| Arxiv | $70.73 \pm 0.35$ | $71.96 \pm 0.18$ | $\mathbf{72.43 \pm 0.20}$ | ↑0.16% |
| Products | $77.17 \pm 0.32$ | $79.01 \pm 0.34$ | $\mathbf{79.17 \pm 0.21}$ | ↑0.18% |

## C.2  VQGRAPH WITH DIFFERENT TEACHER GNNS

In VQGRAPH, we utilize GraphSAGE with GCN aggregation to represent our teacher GNN. However, given that different GNN architectures may exhibit varying performance across datasets, we seek to investigate whether VQGRAPH can perform well when trained with other GNN architectures. As displayed in Figure 7, we present the average performance of VQGRAPH when distilled from different teacher GNNs, including GCN, GAT, and APPNP, across five benchmark datasets. Our results demonstrate that all four teacher models achieved comparable performance, but VQGRAPH consistently outperforms teacher GNNs and other GNN-to-MLP distillation methods, indicating the superior effectiveness and generalization ability of our VQGRAPH.

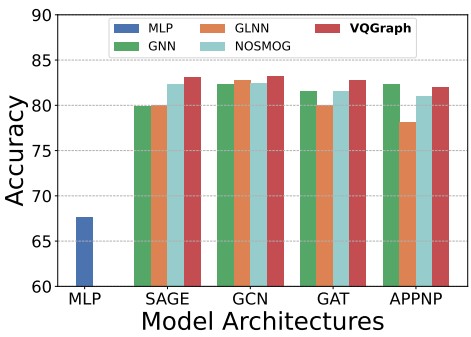

Figure 7: Accuracy vs. Teacher GNN Model Architectures.

## C.3  ROBUSTNESS EVALUATION WITH NOISY FEATURES

We evaluate the robustness of VQGRAPH with regards to different noise levels across different datasets and compute average accuracy. Specifically, we follow Tian et al. (2023b) to initialize the node features and introduce different levels of Gaussian noises to the node features by modifying $X$ with $\tilde{X} = (1-\alpha) \cdot X + \alpha \cdot n$, where $n$ denotes an independent Gaussian noise and $\alpha \in [0, 1]$ controls the noise level. Our results, as illustrated in Figure 8, reveal that VQGRAPH achieves comparable or improved performance compared to GNNs and previous SOTA NOSMOG across different levels

of noise, demonstrating its superior noise-robustness and efficacy, especially when GNNs leverage local structure information of subgraphs to mitigate noise impact. Conversely, GLNN and MLP exhibit rapid performance deterioration as $\alpha$ increases. In the extreme case where $\alpha$ equals 1, and the input features are entirely perturbed, resulting in $\tilde{X}$ and $X$ being independent, VQGRAPH still maintains similar performance to GNNs, while GLNN and MLP perform poorly. Furthermore, even when compared with NOSMOG, which leverages adversarial feature augmentation to combat noise and improve robustness, VQGRAPH still outperforms NOSMOG in terms of noise robustness and overall performance across various $\alpha$ settings, due to its superior ability to sufficiently preserve and leverage graph structural information.

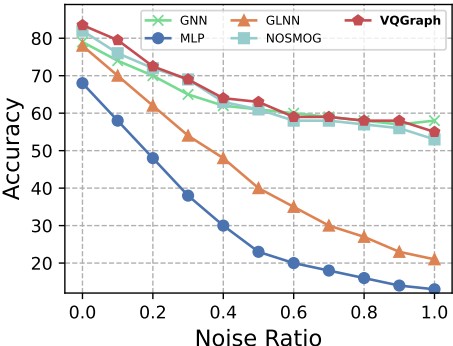

Figure 8: Accuracy vs. Feature Noises.

