# OpenReview forum: "VQGraph: Rethinking Graph Representation Space for Bridging GNNs and MLPs"
_ICLR.cc/2024/Conference — ICLR 2024 poster_

### Official Review · Reviewer_T74P · 2023-10-28

**Soundness:** 3 good
**Presentation:** 3 good
**Contribution:** 2 fair
**Rating:** 5
**Confidence:** 4

**Summary:**

The goal of this paper is to distill the knowledge of "teacher" graph neural networks to "student" MLPs for efficiency. Existing knowledge distillation methods for graph-structured data exploit labels of each node for the knowledge distillation. But, the number of labeled nodes is limited so that the class space may not be sufficient to cover the diverse graph structures. To address this issue, the paper proposes a variant of VQ-VAE to directly label the local graph structures. Specifically, it first learns a structure-aware tokenizer for dealing with graph structured datsets. The structure-aware tokenizer encodes the local substructure of each node as a discrete code. Then, the paper designs a new distillation target to directly transfer the structural knowledge of each node from GNN to MLP.

**Strengths:**

- The paper deals with one of the important research topics on graph representation learning.
- The proposed VQ-VAE is effective and efficient on various datasets and tasks from the author's experiments.
- The paper is well written and easy to follow.

**Weaknesses:**

- It would be better if more details about the production scenario were included in the paper. If the authors are not experts in the graph knowledge distillation domain, they are unlikely to know the details of the experiments. So, more details about the tasks need to be described.
- I wonder whether the proposed VQ-VAE pre-trains both graph tokenizer and graph neural networks simultaneously or pre-trains graph neural networks first and then the graph tokenizer.
- The paper claimed that capturing graph structure is important for knowledge distillation. Then, I think that learning representations with self-supervised learning [1] for graphs can be other option to capture local graph structure. In my thoughts, labeling nodes' local structure with discrete codes seems not different from learning node representation with self-supervision loss [1] if its purpose is capturing local structure.

[1]: Veličković, Petar, et al. "Deep graph infomax." ICLR 2019.

**Questions:**

Please refer to the above weaknesses.

---

> ### Author Response · Authors · 2023-11-17
> **Response to Reviewer T74P**
>
> *We sincerely thank you for your time and efforts in reviewing our paper, and your valuable feekback. We are glad to see that the research topic is important, the proposed method is effective and efficient, and the paper is well written and easy to follow. Please see below for our responses to your comments, and **the revised texts are denoted in red**.*
>
> **Q1: It would be better if more details about the production scenario were included in the paper. If the authors are not experts in the graph knowledge distillation domain, they are unlikely to know the details of the experiments. So, more details about the tasks need to be described.**
>
> A1: Thanks for your constructive suggestion, we have added the following details into **the appendix of the updated manuscript denoted in red**. Regarding to the production scenario, we recognize that real-world deployment of models necessitates the ability to generate predictions for new data points as well as reliably maintain performance on existing ones. Therefore, we apply our method in a realistic production setting to better understand the effectiveness of GNN-to-MLP models, encompassing both transductive and inductive predictions.
>
> In a real-life production environment, it is common for a model to be retrained periodically. The holdout nodes in the inductive set represent new nodes that have entered the graph between two training periods. To mitigate potential randomness and to evaluate generalizability more effectively, we employ a test dataset $V^U_{ind}$, which contains 20% of the test data, and another dataset $V^U_{obs}$, containing the remaining 80% of the test data. This setup allows us to evaluate the model's performance on both observed unlabeled nodes (transductive prediction) and newly introduced nodes (inductive prediction), reflecting real-world inference scenarios.
>
> We further provide three sets of results: "tran" refers to results on $V^U_{obs}$, "ind" refers to $V^U_{ind}$, and "prod" refers to an interpolated value between tran and ind, according to the split rate, indicating the performance in real production settings.
>
> **Q2: I wonder whether the proposed VQ-VAE pre-trains both graph tokenizer and graph neural networks simultaneously or pre-trains graph neural networks first and then the graph tokenizer.**
>
> A2: In our proposed VQ-VAE, both the graph tokenizer and graph neural networks are trained simultaneously, because we need to sufficiently align the feature space between the GNN encoder and the graph tokenizer. By co-training these components, we can ensure a more synchronized enhancement of both the encoder and tokenizer, which can well preserve graph structures. Opting for a sequential approach, where graph neural networks are pre-trained first and then followed by the tokenizer, their inconsistent optimization would potentially lead to suboptimal performance. To further prove this, we conduct an experiment (GraphSAGE as encoder) as follows:
>
> | Method |Simultaneously (Ours)| Sequentially|
> | :-----| :----: | :----: |
> |Citeseer | **70.49** | 70.14 |
> |Pubmed | **75.56** | 75.07 |
> |Cora|**80.64**|80.25|
>
> **Q3: The paper claimed that capturing graph structure is important for knowledge distillation. Then, I think that learning representations with self-supervised learning [1] for graphs can be other option to capture local graph structure. In my thoughts, labeling nodes' local structure with discrete codes seems not different from learning node representation with self-supervision loss if its purpose is capturing local structure.**
>
> A3: Thanks for your insightful comment. We seriously consider your proposed alternative approach for capturing local structures, and find that self-supervised learning methods like Graph Infomax mainly focus on how to recognize local structures and distinguish different local structures, which are not designed for maximally preserving these local structures. Thus, they can not easily recover the whole structures like our graph VQ-VAE, but such knowledge completeness about local structures is critical for GNN-to-MLP distillation. To further prove this, we conduct an experiment (GraphSAGE as encoder) in the table below (we respectively use our VQ-VAE and Graph InfoMax to train a teacher GNN for distillation). From the results, we can find that our method can better transfer the structure-aware knowledge from GNN to MLP, substantially outperforming Graph InfoMax.
>
> | Method |Our Graph VQ-VAE| Graph InfoMax|
> | :-----| :----: | :----: |
> |Citeseer | **76.08** | 73.17 |
> |Pubmed | **78.40** | 76.28 |
> |Cora|**83.93**|81.43|

---

> ### Author Response · Authors · 2023-11-20
> **Gentle Reminder**
>
> Dear Esteemed Reviewer T74P,
>
> We sincerely appreciate the time and effort you dedicated to reviewing our paper. Your thoughtful questions and insightful feedback have been extremely beneficial. In response to your queries and concerns, we have prepared detailed answers .
>
> We understand that you have numerous commitments, and we truly appreciate the time you invest in our work. As the discussion period is approaching its conclusion in one day, we kindly request, if possible, that you review our rebuttal at your earliest convenience.
>
> Should there be any further points that require clarification or improvement, please know that we are fully committed to addressing them promptly.
>
> Thank you once again for your invaluable contribution to our research.
>
> Warm Regards, The Authors

---

> ### Author Response · Authors · 2023-11-23
>
> Dear Reviewer T74P:
>
> We just want to reach out to you again and see if our response addresses your concern. Your comments really inspire us, and we are eager to continue discussing our work with you.
>
> Warm Regards,
>
> The Authors

---

### Official Review · Reviewer_4K7i · 2023-10-31

**Soundness:** 3 good
**Presentation:** 3 good
**Contribution:** 3 good
**Rating:** 8
**Confidence:** 4

**Summary:**

This paper studies the GNN-to-MLP distillation problem. The authors claim that the output representation of teacher GNN lack rich graph structural information. With this motivation, they propose to learn codebook embedding via VQ-VAE. After that, the prediction of teacher and student network is obtained by computing the distance between representation to each of the embedding in the learned codebook. Experimental results show that their methods possess faster inference speed and better performance against previous approaches.

**Strengths:**

- The idea of using VQ-VAE to learn structure-aware embedding is reasonable, which could also serve as an effective approach to extract graph structural feature. Furthermore, the proposed distillation target essentially encourages the representation of each node given by teacher and student network close to the same embedding in the codebook. Therefore, the overall process is more transparent and interpretable.
- This paper is well-organized and easy to follow. The experiments results are convincing and comprehensive, which are enough to support the effectiveness of the proposed method.

**Weaknesses:**

- Compared with previous method, extra memory space is required due to the introduced learnable codebook. As the authors show, the codebook size has influence on the performance of the model. How to effective tune this hyperparameter is worth exploring.
- The adopted datasets in this paper are generally homophilic, i.e., the connected nodes belong to the same category. It is still unclear whether the proposed method is applicable to heterophilic graphs.

**Questions:**

- The authors propose a generative approach learn codebook embedding. Is there any other way to learn the codebooks (such as self-supervised learning paradigm) or use other generative models (such as GAN) ?
- Could the proposed method be applied to heterophilic graphs [1,2]?

[1]  Lim et al., Large scale learning on non-homophilous graphs. NeurIPS 2021.

[2] Platonov et al, A critical look at the evaluation of GNNs under heterophily: Are we really making progress? ICLR 2023.

---

> ### Author Response · Authors · 2023-11-17
> **Response to Reviewer 4K7i**
>
> *We sincerely thank you for your time and efforts in reviewing our paper, and your valuable feekback. We are glad to see that the idea is reasonable, the overall process is more transparent and interpretable, this paper is well-organized and easy to follow, and the experiment results are convincing and comprehensive. Please see below for our responses to your comments, and **the revised texts are denoted in red**.*
>
> **Q1: Compared with previous method, extra memory space is required due to the introduced learnable codebook. As the authors show, the codebook size has influence on the performance of the model. How to effective tune this hyperparameter is worth exploring.**
>
> A1: The size of the codebook inherently possesses an optimal point, a fact substantiated by our experimental findings. Notably, this optimum tends to scale **proportionally with the growth in the number of nodes** and features within a graph. To address practical concerns, we propose an efficient approach involving **a few key steps**. Specifically, employing a binary search process, we can determine the optimal codebook size within 3-4 steps, where it can be determined **solely by convergence** without running the entire training process, especially in large-scale datasets. Meanwhile, minor adjustments to the codebook size, especially within a range not comparable with the scale of graph complexity, have negligible impact on the overall performance. This streamlined process ensures that adapting the codebook size to new datasets is both manageable and expeditious.
>
> **Q2: The adopted datasets in this paper are generally homophilic, i.e., the connected nodes belong to the same category. It is still unclear whether the proposed method is applicable to heterophilic graphs.**
>
> A2: Capturing local structures is also crucial for heterophilic graphs, thus we apply our method to heterophilic graphs. Specifically, we employ two state-of-the-art Graph Neural Network (GNN) models known for their efficacy on heterophilic graphs, namely ACMGCN and GCNII, along with MLP. Incorporating our structure-aware code embeddings, we evaluate these models on two challenge datasets—Texas and Cornell.
> The results of these experiments, displayed below, demonstrate that our graph VQ-VAE **can generalize to different heterophilic GNN/MLP architectures and consistently improve their performances**. These remarkable results underscore the effectiveness of our method in effectively capturing local structural information, and demonstrate the versatility and broad applicability of our proposed approach across different graph domains. **We have added this part into the updated manuscript.**
>
> | Dataset | ACMGCN | ACMGCN with Graph VQ-VAE | MLP   | MLP with Graph VQ-VAE | GCNII | GCNII with Graph VQ-VAE |
> | :-----| :----: | :-----:| :----: | :-----:|  :-----:| :----: |
> | texas   | 86.49  | **87.03**      | 75.68 | **77.93** | 76.73 | **79.04**    |
> | cornell | 84.05  | **84.59**          | 76.38 | **78.29** | 76.49 | **78.29**    |
>
> **Q3: The authors propose a generative approach learn codebook embedding. Is there any other way to learn the codebooks (such as self-supervised learning paradigm) or use other generative models (such as GAN) ?**
>
> A3: Any self-supervised learning or generative modeling methods that use graph reconstruction objective can learn a structure-aware codebook for graph data. However, self-supervised learning methods with mask-prediction can **only recover partial structures** which are not expressive enough for a structure-aware codebook. Other generative models like GAN pay more attention to how to learn **a good decoder not a good encoder (codebook)**. In contrast to them, our graph VQ-VAE effectively combine  representation learning with generative modeling for learning a expressive codebook that maximally preserves the entire graph structures.

---

> ### Author Response · Authors · 2023-11-20
> **Gentle Reminder**
>
> Dear Esteemed Reviewer 4K7i,
>
> We sincerely appreciate the time and effort you dedicated to reviewing our paper. Your thoughtful questions and insightful feedback have been extremely beneficial. In response to your queries and concerns, we have prepared detailed answers .
>
> We understand that you have numerous commitments, and we truly appreciate the time you invest in our work. As the discussion period is approaching its conclusion in one day, we kindly request, if possible, that you review our rebuttal at your earliest convenience.
>
> Should there be any further points that require clarification or improvement, please know that we are fully committed to addressing them promptly.
>
> Thank you once again for your invaluable contribution to our research.
>
> Warm Regards, The Authors

---

> ### Comment · Reviewer_4K7i · 2023-11-22
> **Response to the Authors**
>
> Thanks the authors for their detail responses. My concerns have been well addressed. Currently I increase my score to 8.

---

> > ### Author Response · Authors · 2023-11-22
> > **Thanks for Your Support**
> >
> > Dear Reviewer，
> >
> > Many thanks for raising score! We sincerely appreciate your valuable comments and your precious time in reviewing our paper!
> >
> > Warm Regards,
> >
> > The Authors

---

### Official Review · Reviewer_F8Tt · 2023-11-07

**Soundness:** 3 good
**Presentation:** 3 good
**Contribution:** 3 good
**Rating:** 8
**Confidence:** 4

**Summary:**

This work proposes a new method of knowledge distillation for node classification. Essentially, VQGraph incorporates a VQ-VAE to learn a codebook that represents informative local structures, and use these local structures as additional information, distilling it to student MLPs. Empirically, this approach outperforms state-of-the-art methods, in both accuracy and inference speed.

**Strengths:**

1. Although VQ-VAE is not a new method, introducing a variant of VQ-VAE for KD for node classification sounds novel, and also make sense.

2. The paper is well-written and easy to follow.

3. This approach is effective under both inductive and transdutive setting, and also works well for large-scale datasets.

4. The experiment is comprehensive and convincing.

**Weaknesses:**

1. The size of codebook is sensitive to different datasets, this may introduce some practical difficulty.

2. It would be better to provide a theoretical understanding of this approach.

**Questions:**

1. It seems that for APPNP, all three KD approaches can't outperform the teacher model. Is there any discussion or explanation for this phenomenon?

2. In Table 4, Only-VQ outperforms Class-based and AE+Class-based, but Only-VQ only adopts class soft labels, the VQ component helps to train the codebook, why this approach is better than the other two? Can you give some discussions on it?

3. It would be better to provide an analysis what codebook entry is most informative to a class label, which can make this approach more convincing and intuitive.

---

> ### Author Response · Authors · 2023-11-17
> **Response to Reviewer F8Tt**
>
> *We sincerely thank you for your time and efforts in reviewing our paper, and your valuable feekback. We are glad to see that the proposed method is novel and makes sense, the paper is  well-written and easy to follow, this approach is effective and the experiment is comprehensive and convincing. Please see below for our responses to your comments, and **the revised texts are denoted in red**.*
>
> **Q1: The size of codebook is sensitive to different datasets, this may introduce some practical difficulty.**
>
> A1: The size of the codebook inherently possesses an optimal point, a fact substantiated by our experimental findings. Notably, this optimum tends to scale **proportionally with the growth in the number of nodes** and features within a graph. To address practical concerns, we propose an efficient approach involving **a few key steps**. Specifically, employing a binary search process, we can determine the optimal codebook size within 3-4 steps, where it can be determined **solely by convergence** without running the entire training process, especially in large-scale datasets. Meanwhile, minor adjustments to the codebook size, especially within a range not comparable with the scale of graph complexity, have negligible impact on the overall performance. This streamlined process ensures that adapting the codebook size to new datasets is both manageable and expeditious.
>
> **Q2: It would be better to provide a theoretical understanding of this approach.**
>
> A2: We provide a theoretical understanding for VQGraph, assuming the data adimts the following hierarchical generation mechanism:
> $$\begin{array}{rl}
>   &c_i \sim q(c_i)\\\\
>   &r_i \sim q(r_i|c_i)\\\\
>   &G \sim q(G|(r_1,r_2,...r_N)), \\\\
> \end{array}$$
> where $c_i,r_i$ are the label and latent variable (code) for the i-th node in the graph $G=(V,A)$. In other words, we assume there exist an explicit latent representation for each node that can be used to generate the observed graph. Importantly, these code representations encode not only feature information but also structure information, so that the adjacency matrix can be generated. VQGraph can be viewed as using generative pre-training to recover the latent representation and facilitate the label prediction, as learning
> $$q(c_i|r_i)$$ is easier than learning $$q(c_1,c_2,...c_N|G) = \int \prod_i q(c_i|r_i)q(r_1,r_2,...r_N|G)dr$$
>
>
> **Q3: It seems that for APPNP, all three KD approaches can't outperform the teacher model. Is there any discussion or explanation for this phenomenon?**
>
> A3: This is due to the fact that the APPNP directly uses node features for prediction prior to the message passing on the graph, which is very similar to what the student MLP does in KD approaches. Therefore using APPNP as teacher model can not provide student MLP with much additional information.
>
> **Q4: It would be better to provide an analysis what codebook entry is most informative to a class label, which can make this approach more convincing and intuitive.**
>
>
>
> A4: In the training stage, we tokenize each node with different neighborhood structures as discrete codes using Graph VQ-VAE. Now we take a closer look at codebook entries for different classes on Pubmed for in-depth analysis. There is little difference in the number of codes with each class and there is a uniform distribution of the nodes for each class, demonstrating the learning capacity of our model. Meanwhile, the results  demonstrate that there is small code overlap among different classes, which indicates the code distributions learned for different classes are mutually independent. This finding validates that our graph VQ-VAE enables distinct separation of the nodes with different class labels in the representation space, facilitating a better knowledge transfer in GNN-to-MLP distillation. We have added these analysis into the updated manuscript denoted in red.
>
> Code Entries for each class of Pubmed:
>
> |              | Class 1 | Class 2 | Class 3 |
> | ------------ | ------- | ------- | ------- |
> | Code Entries | 2490    | 2623    | 2591    |
>
> Number of overlapping codes for each pair of classes:
>
> |         | Class 1 | Class 2 | Class 3 |
> | ------- | ------- | ------- | ------- |
> | Class 1 | *       | 24      | 21      |
> | Class 2 |         | *       | 19      |
> | Class 3 |         |         | *       |

---

> ### Author Response · Authors · 2023-11-20
> **Gentle Reminder**
>
> Dear Esteemed Reviewer F8Tt,
>
> We sincerely appreciate the time and effort you dedicated to reviewing our paper. Your thoughtful questions and insightful feedback have been extremely beneficial. In response to your queries and concerns, we have prepared detailed answers .
>
> We understand that you have numerous commitments, and we truly appreciate the time you invest in our work. As the discussion period is approaching its conclusion in three days, we kindly request, if possible, that you review our rebuttal at your earliest convenience.
>
> Should there be any further points that require clarification or improvement, please know that we are fully committed to addressing them promptly.
>
> Thank you once again for your invaluable contribution to our research.
>
> Warm Regards, The Authors

---

> > ### Comment · Reviewer_F8Tt · 2023-11-20
> >
> > Thank you for you detailed reply. I have some questions regarding your reply:
> >
> > 1) Regarding the search procedure in Q1, you said that "we can determine the optimal codebook size within 3-4 steps", can you explain it more specific. My understandingis that you will start with some heuristic value, e.g., 128, then increase the size by a factor, say 2, then check the performance in validation set. Please correct me if I am wrong.
> >
> > 2) Also in Q1, "solely by convergence without running the entire training process", does it mean you would sample a subset of training data, entire training process means entire training dataset?
> >
> > 3) In Q4, can you please elaborate how do you define code overlap and code distributions?

---

> > > ### Author Response · Authors · 2023-11-20
> > > **Response to Reviewer F8Tt**
> > >
> > > Thanks for your quick and kind comments, we here answer your further questions:
> > >
> > > **For Q1**: Our selection process can be specified as follows. Taking the Citeseer dataset as an example, which includes 2110 nodes, we initially choose a value around the number of nodes, i.e., 2048. We then test the performance of this codebook size, as well as two other sizes, one double and the other quadruple, on the validation set. We plot a curve based on this performance. If the curve exhibits a parabolic shape, indicating that the performance is optimal at double the size, we designate that as the optimal codebook size, which is just the case on Citeseer. If the curve shows a monotonically decreasing or increasing trend, we continue either halving the smallest codebook size or doubling the largest codebook size. This process is repeated until the optimal point is found. If a more precise size is desired, a binary search can be conducted, but our experimental results indicate that there is not much difference in performance within a smaller range near the peak. Using this method on different datasets, we find our codebook size with an average of 3-4 testing steps.
> > >
> > > **For Q2**: Yes, we would randomly sample a subset of training data to conduct preliminary codebook selection (in above answer), and then use the selected codebook size for the whole training.
> > >
> > > **For Q3**: After the training of graph tokenizer, we assign each node to a token ID, and we can obtain a cumulative distribution over all token IDs (**code distributions**) of each class' nodes. If the nodes with different classes are assigned with a same token ID, it would be recognized as one **code overlap**.
> > >
> > > Should there be any further points that require clarification or improvement, please know that we are fully committed to addressing them promptly.
> > >
> > > Thank you once again for your invaluable contribution to our research.
> > >
> > > Warm Regards,
> > >
> > > The Authors

---

> ### Comment · Reviewer_F8Tt · 2023-11-20
>
> 1. "Taking the Citeseer dataset as an example, which includes 2110 nodes, we initially choose a value around the number of nodes, i.e., 2048". But if the dataset contains hundreds of thousands nodes or even millions of nodes, would this approach consume a lot of memory, or you have other method to determine the size of the codebook?

---

> > ### Author Response · Authors · 2023-11-20
> > **Response to Reviewer F8Tt**
> >
> > Regarding the size selection in large-scale dataset, we would try different codebook size with exponential growth, such as $2^{12}->2^{13}$. Usually, we do not need to use the codebook size as large as nodes, such as the experiment we conducted on Arxiv dataset. Most importantly, we **do not introduce additional inference costs into student MLP** because the codebook is only learned for GNN-to-MLP distillation.
> >
> > Thank you once again for your invaluable contribution to our research.
> >
> > Warm Regards,
> >
> > The Authors

---

> > > ### Comment · Reviewer_F8Tt · 2023-11-21
> > >
> > > Thank you for your clear response. Most of my concerns have been addressed. I think it is a solid work, and I raise my score to 8.

---

> > > > ### Author Response · Authors · 2023-11-21
> > > > **Thanks for Your Support**
> > > >
> > > > Many thanks again for raising score! We sincerely appreciate your valuable comments and your precious time and efforts in reviewing our paper!
> > > >
> > > > Warm Regards,
> > > >
> > > > The Authors

---

### Official Review · Reviewer_Cnqf · 2023-11-08

**Soundness:** 2 fair
**Presentation:** 3 good
**Contribution:** 2 fair
**Rating:** 6
**Confidence:** 4

**Summary:**

This paper presents a novel KD method from GNNs to MLPs to achieve the improvement of inference efficiency, where a special codebook is designed to store the graph information. Experiments on a wide range of benchmarks show its superior performance in terms of both accuracy and time cost.

**Strengths:**

- How to make MLP more aware of the graph structure is a valuable topic for better inference performance. (while is also not that addressed in this paper, see the first weakness and question below)
- The whole paper is well structured, the explanation for each part is overall complete and clear to understand.
- The experimental results are convincing, indeed providing many interesting observations for discussion (see questions below).

**Weaknesses:**

- The connection between GNNs-to-MLPs distillation and the proposal of a local-structure-aware codebook is less developed, since they are actually separate goals; more explanation is needed to make the main claim of the paper clearer.
- In practice, one of the main concerns is that the hyper-parameter for the code distillation in equation 7 is quite small compared to the class distillation one. It makes the whole proposal more intended to be a node embedding booster for GNNs, the graph tokenizer, while the codebook is not significantly effective in the distillation phase.
- The volume of the codebook is very likely to be the same scale as the original node features, which makes the proposal still in the trade-off of time and space cost for sure; therefore, more information about the parameter volume should be provided.

**Questions:**

- Major
    - In the abstract, the authors mention that 'the class space may not be expressive enough to capture numerous different local graph structures'. I am somehow confused, since it is about GNNs-to-MLPs KD, better capturing the local structure information is an issue for GNNs part, instead of KD?
    - It is better to explain more about why it is appropriate to use VQ-VAE to encode graph structure, since it was originally designed for continuous data. For graph-structured data, it might be easier to convert to frequency information, i.e., each position in the "codebook" stores the volume of a specific frequency. I am just curious why VQ-VAE is suitable for encoding discrete and non-Euclidean data.
    - According to the inner product in the reconstruction loss of equation 3 and the benchmarks used, it is quite important to emphasize that the effective domain of the proposal is probably homophilic graphs, otherwise further discussion or experiments on heterophilic graphs are necessary.

- Minor
    - What is the data set used for Figure 1? Cora? All other figures/tables need further clarification on which dataset is used.
    - In equation 3, the node embeddings are still included, so the claimed local structural embedding is not that promising, can you compare or explain the difference of the two parts of the reconstruction function? Also, what's the nonlinear function here?
    - Figure 2 needs improvement- In equation 7, the loss of classification and class distillation should also be clarified.
    - The codebook does not seem to be so different from VGAE from the point of view of structure reconstruction, but the node feature reconstruction part makes it so different. Also, the graph structure part is not well designed, but an inner product.
    - In the figures and descriptions throughout the paper, it is not fair to say that the codebook really preserves the structural information as a substructure, since it is just another integrated embedding where the identifiability of specific structure is missing, therefore it is misleading to give the local structure in Figure 2.
    - An interesting point, which is worth to study further, is what the Figure 5(ii) implies: denser graph needs more space, which is parallel to the discovery from graph signal processing domain, that the structural information cannot be summarized only by some low-frequency filters, the high-frequency ones are also informative in such graphs.

---

> ### Author Response · Authors · 2023-11-17
> **Response to Reviewer Cnqf (Part 1/3)**
>
> *We sincerely thank you for your time and efforts in reviewing our paper, and your valuable feekback. We are glad to see that the proposed method is novel, the whole paper is well structured, the explanation is clear to understand, and the experimental results are convincing with many interesting observations. Please see below for our responses to your comments, and **the revised texts are denoted in red**.*
>
> **Q1:The connection between GNNs-to-MLPs distillation and the proposal of a local-structure-aware codebook is less developed, since they are actually separate goals; more explanation is needed to make the main claim of the paper clearer.**
>
> A1: We here clarify the connections between GNNs-to-MLPs distillation and the proposition of a local-structure-aware codebook. The primary goal of GNNs is to establish an embedding space encompassing all nodes, wherein structural information is consolidated across multiple layers of graph convolutional layers. Kindly recall that the aim of GNN-to-MLP distillation is to facilitate the transfer of sufficient structure-based information to the MLP, enabling the latter to contain graph-related information. Unlike GNNs, MLPs are inherently limited in their capacity to capture structural information independently. Therefore, the knowledge distillation process serves for transferring local structural information to the MLP. The integration (a codebook) of these local-structure-aware codes work as an explicit embedding space for storing and conveying this local structural information, thereby enhancing the MLP's capacity to acquire such knowledge.
>
> **Q2: In practice, one of the main concerns is that the hyper-parameter for the code distillation in equation 7 is quite small compared to the class distillation one. It makes the whole proposal more intended to be a node embedding booster for GNNs, the graph tokenizer, while the codebook is not significantly effective in the distillation phase.**
>
> A2: Regarding the disparity in hyperparameters between the code distillation and class distillation in Equation 7, it is critical to consider the calculation processes for both the class-based distillation loss and code-based distillation loss. The class-based distillation loss entails computing the KL divergence between the student predictions and soft labels. On the other hand, the code-based loss involves comparing the target node representations with all M codes of the codebook embeddings.
> It is deserved to notice that **the class number and code number are extremely different regarding the order of magnitude**, which results in a considerable scale gap of approximately 10^7-fold in large-scale datasets. Consequently, despite the smaller hyperparameter, the corresponding code distillation loss is proportionally larger, effectively ensuring that **the gradients** propagated during backpropagation are maintained **at a comparable level with class distillation loss**.
>
> **Q3: The volume of the codebook is very likely to be the same scale as the original node features, which makes the proposal still in the trade-off of time and space cost for sure; therefore, more information about the parameter volume should be provided.**
>
> A3: Indeed, the trade-off between time and space costs is a long-term challenge in designing efficient graph neural network architectures. From this perspective, our proposed VQGraph is able to effectively capture the structural diversity of graph data with a compact codebook. Our method focuses on achieving a balance where the size of the codebook is sufficiently large to encapsulate the essential substructure patterns yet remains much smaller than the original substructures in graph data. The codebook’s volume is not directly proportional to the original node features but is mainly determined by the complexity of the graph data, considering both nodes and edges which produce different local substructures. Taking Cora dataset for example, while the **codebook size is 2048**, this allows us to efficiently represent a manifold larger number of possible 1-hop substructures. Considering Cora’s 2485 nodes with an average degree of about 4, there is **a theoretical limit of ( O(2485^4) ) distinct 1-hop substructure patterns**. Thus, our codebook provides a highly compressed representation that encapsulates these patterns with a significantly lower memory footprint.
> Moreover, the empirical performance of our method demonstrates that despite the compact size of the codebook, there is no compromise on the expressiveness and the ability to facilitate meaningful representations. We want to highlight that the codebook size represents a new effective embedding space that strikes a balance between memory efficiency and the capacity to represent diverse structural information.

---

> ### Author Response · Authors · 2023-11-17
> **Response to Reviewer Cnqf (Part 2/3)**
>
> **Q4: It is better to explain more about why it is appropriate to use VQ-VAE to encode graph structure, since it was originally designed for continuous data. For graph-structured data, it might be easier to convert to frequency information, i.e., each position in the "codebook" stores the volume of a specific frequency. I am just curious why VQ-VAE is suitable for encoding discrete and non-Euclidean data.**
>
> A4: Whether it is for continuous data or for discrete data, the core of the encoder is to learn a optimal embedding space for maximally preserving the intrinsic information of the raw data, which is also the primary goal of VQ-VAE. Thus, the utilization of VQ-VAE is essentially not related to the data format. Besides, the discrete nature of graph-structured data aligns with the VQ-VAE's ability to handle categorical and non-continuous information with indexed codes. Each position in the codebook can encapsulate essential structural information, thereby facilitating the encoding of complex graph structures.
>
>
> **Q5: According to the inner product in the reconstruction loss of equation 3 and the benchmarks used, it is quite important to emphasize that the effective domain of the proposal is probably homophilic graphs, otherwise further discussion or experiments on heterophilic graphs are necessary.**
>
> A5: Capturing local structures is also crucial for heterophilic graphs, thus we apply our method to heterophilic graphs. Specifically, we employ two state-of-the-art Graph Neural Network (GNN) models known for their efficacy on heterophilic graphs, namely ACMGCN and GCNII, along with MLP. Incorporating our structure-aware code embeddings, we evaluate these models on two challenge datasets—Texas and Cornell.
> The results of these experiments, displayed below, demonstrate that our graph VQ-VAE **can generalize to different heterophilic GNN/MLP architectures and  consistently improve their performances**. These remarkable results underscore the effectiveness of our method in effectively capturing local structural information, and demonstrate the versatility and broad applicability of our proposed approach across different graph domains. Moreover, We want to emphasize that the inner product is **specifically for measuring the relationships between neighboring nodes, not specifically for homophilic graphs**. And it can be replaced by any non-parametric or parameterized relation measurements. **We have added this part into the updated manuscript.**
>
> | Dataset | ACMGCN | ACMGCN with Graph VQ-VAE | MLP   | MLP with Graph VQ-VAE | GCNII | GCNII with Graph VQ-VAE |
> | :-----| :----: | :-----:| :----: | :-----:|  :-----:| :----: |
> | texas   | 86.49  | **87.03**      | 75.68 | **77.93** | 76.73 | **79.04**    |
> | cornell | 84.05  | **84.59**          | 76.38 | **78.29** | 76.49 | **78.29**    |
>
> **Q6: What is the data set used for Figure 1? Cora? All other figures/tables need further clarification on which dataset is used.**
>
> A6: We appreciate your attention to detail and apologize for any confusion. The dataset employed for Figure 1 and Figure 4 is Citeseer. In our paper, we have provided the datasets used for other figures and tables.
>
> **Q7: In equation 3, the node embeddings are still included, so the claimed local structural embedding is not that promising, can you compare or explain the difference of the two parts of the reconstruction function? Also, what's the nonlinear function here?**
>
> A7: Certainly. In the early stages of our experiments, we conducted comprehensive ablation studies on the distinct components of our reconstruction function. The result is presented in the following table. From the result, it is obvious that the edge reconstruction segment is significantly better than the node embeddings in terms of its contribution to the overall performance. While the node embeddings does lead to a certain improvement in results, we include them in our reconstruction loss for better results. The nonlinear function here is the sigmoid function.
>
> | Dataset    | Only node reconstruction | Only edge reconstruction | node reconstruction+edge reconstruction |
> | :----: | :----: | :----: | :----: |
> | Citeseer   | 74.84                    | 75.56                    | 76.09                                   |
> | Pubmed     | 77.52                    | 78.10                    | 78.40                                   |
> | Cora       | 83.15                    | 83.72                    | 83.93                                   |
> | A-computer | 84.16                    | 85.03                    | 85.17                                   |
> | A-photo    | 93.63                    | 94.15                    | 94.21                                   |
> | Arxiv      | 71.17                    | 72.19                    | 72.43                                   |
> | Products   | 78.40                    | 78.84                    | 79.17                                   |

---

> ### Author Response · Authors · 2023-11-17
> **Response to Reviewer Cnqf (Part 3/3)**
>
> **Q8: Figure 2 needs improvement- In equation 7, the loss of classification and class distillation should also be clarified.**
>
> A8: We have updated Figure 2 in the updated manuscript according to your suggestions.
> $L_{cls}$ is the cross-entropy loss between the MLP prediction $\hat y_{{v}
> }$ and the class label $y_{v}$.
> $L_{labeldistill}$ is the KL-divergence loss between the student prediction $\hat y_{v}$ and the soft label $y^{soft}_{v}$
>
> **Q9: The codebook does not seem to be so different from VGAE from the point of view of structure reconstruction, but the node feature reconstruction part makes it so different. Also, the graph structure part is not well designed, but an inner product.**
>
> A9: The critical difference between our graph VQ-VAE and VGAE is that VQ-VAE utilizes an explicit latent space (codebook) for enhancing the structure-aware graph representation learning while VGAE learns an implict latent space for graph generative modeling. The former focuses on learning a better graph encoder and the latter pays more attention to a better graph decoder (generator). From this perspective, it is reasonable to use a somple inner product as graph decoder in training our graph VQ-VAE, because a well-designed decoder would make the model to **depend heavily on the decoder** to reconstruct graph structures, thus limiting the sufficient learning of graph encoder.
>
>
> **Q10: An interesting point, which is worth to study further, is what the Figure 5(ii) implies: denser graph needs more space, which is parallel to the discovery from graph signal processing domain, that the structural information cannot be summarized only by some low-frequency filters, the high-frequency ones are also informative in such graphs.**
>
> A10: We are delighted to discover that our findings align with those in the graph signal processing domain. Indeed, low-frequency filters excel at capturing smooth or slow-changing signal shifts, while high-frequency filters are informative for capturing fast-changing signal shifts within a graph.In real-life production scenarios, incorporating these high-frequency filters with graph signal learning can greatly enrich the representation of local structures, which are often diverse in large graphs.
>
> To further extend our framework, we believe it can be applied to other domains as well. For instance, we can employ a 2-way tokenizer to encode the local shifting structures into codebooks. This constraint serves as a regularization function, guiding the model to focus on the "most important" substructure information that best reconstructs the graph. In this approach, we could utilize separate codebooks for low-frequency and high-frequency filters, allowing for different codebook sizes. We assign a smaller codebook to low-frequency filters to facilitate overall and smooth learning, while allocating a larger codebook to high-frequency filters to capture more detailed and localized information. By employing this strategy, we aim to capture both the local and global structural information embedded within graph signals.
>
> In summary, our framework not only aligns with insights from the graph signal processing domain but also presents opportunities for further exploration. By incorporating a 2-way tokenizer and utilizing separate codebooks for low- and high-frequency filters, we can enhance the representation of both global and local structures in various domains.

---

> ### Author Response · Authors · 2023-11-20
> **Gentle Reminder**
>
> Dear Esteemed Reviewer Cnqf,
>
>
> We sincerely appreciate the time and effort you dedicated to reviewing our paper. Your thoughtful questions and insightful feedback have been extremely beneficial. In response to your queries and concerns, we have prepared detailed answers .
>
> We understand that you have numerous commitments, and we truly appreciate the time you invest in our work. As the discussion period is approaching its conclusion in one day, we kindly request, if possible, that you review our rebuttal at your earliest convenience.
>
> Should there be any further points that require clarification or improvement, please know that we are fully committed to addressing them promptly.
>
> Thank you once again for your invaluable contribution to our research.
>
> Warm Regards,
> The Authors

---

> > ### Comment · Reviewer_Cnqf · 2023-11-21
> > **Response**
> >
> > Dear all authors,
> >
> > Thank you for your comprehensive response to my concerns, especially the details about the parameters, the heterophilicity experiments, and the comparison of edge/node reconfiguration contributions. However, my concerns with this paper remain twofold:
> >
> > - First, the motivation of the codebook does not seem to be directly related to the purpose of KD, as it is not specialized in extracting information from GNNs-to-MLPs but more from graphs to representation models. So, the proposed methodology is actually a scheme to enhance the GNN model, rather than specifically targeting the KD part; it is somehow disconnected from the title.
> > - The connection between VQ-VAE and graph structure representation lacks further explanation, especially the intuition behind it, i.e., how a piece of code in the codebook relates to the graph structure, and it would be desirable to have some empirical results to help the reader understand this connection and to make the approach more rational. As it stands, the explanations provided from the perspective of representation learning are too general.
> >
> > Based upon the above questions are not appropriately aggressed, I would rather keep my score. The author's follow-up response will also be considered.
> >
> > Best,

---

> > > ### Author Response · Authors · 2023-11-22
> > > **Response to Reviewer Cnqf**
> > >
> > > Thanks for your reply, we further response to your concerns as follows.
> > >
> > > **Q1: The motivation of the codebook does not seem to be directly related to the purpose of KD, as it is not specialized in extracting information from GNNs-to-MLPs but more from graphs to representation models.**
> > >
> > > A1: The most important point in GNN-to-MLP distillation is how to transfer the **structure-aware** node representations from GNN to MLP. Please note that previous methods learn teacher GNN **mainly by class-based objective**, and their leanred node representations only **utilize structure not preserve structure**. Thus the structural expressiveness of teacher GNN representations can be the limitation of current GNN-to-MLP distillation methods. A more straightforward explanation has been illustrated in Figure 1. Then the problem is how to maximally preserve local structures in teacher GNN and make more efficient structure-based GNN-to-MLP distillation. The codebook learned by our graph VQ-VAE is proven to be an effective structure "carrier" that well preserves diverse local graph structures, which involves **structure-based reconstruction objective**. Based on the codebook, our **code-based (structure-based) distillation objective** further improves the distillation results as demonstrated in the ablation study of **Table 4, where *VQ+code-based objective* substantially improves *only-VQ* in distillation results**.
> > > In conclusion, the codebook improves GNN-to-MLP methods from two aspects:
> > >
> > > * First, it makes the transferred node representations more expressive in representing local graph structures in **GNN pre-training stage** because a better teacher is critical for knowledge distillation [1].
> > >
> > > * Second, it contributes to the new powerful structure-based distillation objective in **GNN-to-MLP distillation stage**, which significantly improve the final distillation results.
> > >
> > > [1] Huang T, You S, Wang F, et al. Knowledge distillation from a stronger teacher[J]. Advances in Neural Information Processing Systems, 2022, 35: 33716-33727.
> > >
> > > **Q2: The connection between VQ-VAE and graph structure representation needs further explanation**
> > >
> > > A2: To better illustrate how a piece of code in the codebook relates to the graph structure, we add empirical analysis **in Figure 6 of the updated manuscript**, where we conduct subgraph retrieval experiment, please kindly check out it. From the results, we can observe that our code (token) ID can precisely identify and reflect the different local graph structures, while previous SOTA method NOSMOG fails to do that, demonstrating the effectiveness of our structure-aware GNN-to-MLP distillation method.
> > >
> > > Should there be any further points that require clarification or improvement, please know that we are fully committed to addressing them promptly.
> > >
> > > Thank you once again for your invaluable contribution to our research.
> > >
> > > Warm Regards,
> > >
> > > The Authors

---

> > > > ### Comment · Reviewer_Cnqf · 2023-11-22
> > > > **Response**
> > > >
> > > > I am grateful to the authors for providing a further explanation, which this time is more helpful in resolving these two issues, especially since the supplementary experiments in Figure 6 provide consistent evidence. However, for the first issue, despite the second point of the explanation certainly answered my question, the two phases remain entangled as the codebook has been necessary.
> > > >
> > > > Anyways, since most of my concerns are addressed, and I would like to raise my score to 6. Good luck!

---

> > > > > ### Author Response · Authors · 2023-11-22
> > > > > **Thanks for Your Support**
> > > > >
> > > > > Many thanks again for raising score! Your further comments and insights would be invaluable to enhance our work. Thank you once again for your invaluable contribution to our research.
> > > > >
> > > > > Warm Regards,
> > > > >
> > > > > The Authors

---

### Official Review · Reviewer_12WL · 2023-11-10

**Soundness:** 2 fair
**Presentation:** 3 good
**Contribution:** 2 fair
**Rating:** 5
**Confidence:** 3

**Summary:**

This paper proposes a GNN-to-MLP distillation approach. During the teacher GNN training, a series of code embeddings are jointly optimized using a reconstruction loss, to make them encode local graph structures. The divergence between the soft code assignment of the teacher GNN and that of the student MLP is used as additional supervision for distillation. The proposed approach outperforms the teacher GNN and other existing GNN-to-MLP distillation methods in both transductive and inductive settings.

**Strengths:**

- The proposed approach is novel compared with existing GNN-to-MLP distillation methods.
- Experimental results are good, consistently better than NOSMOG
- Additional ablation studies

**Weaknesses:**

- Some details of the experimental setup are not clear enough to me. As far as I understand from the paper, in the inductive setting,  $G^L \cup G^U_{obs}$ is used for training the teacher GNN and for distillation. True labels $Y^L$ are used to train the teacher but are not used in distillation. Is it right? The paper mentions that "the edges between $G^L \cup G^U_{obs}$ and $G^U_{ind}$ are removed in training, while they are leveraged during inference to transfer positional features via average operator". What does this mean? Do you augment the node feature with the features of its neighbors for inference?
- What the student MLP can learn from distillation is ambiguous. Suppose two nodes have the same private feature but different neighborhoods, the teacher GNN would generate different representations and different soft code assignments for these two nodes. However, in distillation, the student MLP would always generate the same representation and soft code assignment for these two nodes (since they have the same private feature). As a result, the student MLP cannot learn their structural difference from the teacher

**Questions:**

Please see Weaknesses

---

> ### Author Response · Authors · 2023-11-17
> **Response to Reviewer 12WL**
>
> *We sincerely thank you for your time and efforts in reviewing our paper, and your valuable feekback. We are glad to see that the proposed method is novel, the experimental results are good, and the ablation studies are adequate. Please see below for our responses to your comments, and **the revised texts are denoted in red**.*
>
> **Q1: Some details of the experimental setup are not clear. Are true labels $Y_L$ used to train the teacher but not used in distillation? Do you augment the node feature with the features of its neighbors for inference?**
>
> A1:  In our inductive experimental settings, we align with the framework established by NOSMOG [1] to ensure a fair comparison.
>
> * To clarify, we utilize true labels $Y_L$ for both training the teacher GNN and for GNN-to-MLP distillation process. In training the student, we employ soft labels and soft code assignments within the labeled and observed subsets to compute the class-based distillation loss and the code-based distillation loss. The classification loss is defined as the cross-entropy loss between the MLP prediction and the true labels $Y_L$.
>
> * Indeed, in the inference process, we adopt the practice of augmenting the node feature with its neighbors' features, in line with the approach outlined in NOSMOG and  DeepWalk. This augmentation strategy aims to provide nodes with additional information, thereby enriching the inferencing capabilities. With this methodology, we aim to circumvent potential limitations, such as the occurrence of nodes with identical features but dissimilar neighbors, a scenario that could pose challenges for the MLP in effectively processing the information. This instance directly addresses the concern raised in your second question, and our elucidation stands as follows.
>
> **Q2: What the student MLP can learn from distillation is ambiguous. Suppose two nodes have the same private feature but different neighborhoods, the teacher GNN would generate different representations and different soft code assignments for these two nodes. However, in distillation, the student MLP would always generate the same representation and soft code assignment for these two nodes (since they have the same private feature). As a result, the student MLP cannot learn their structural difference from the teacher**
>
> A2: It is true that in distillation, the student MLP, based solely on node features, may not fully capture the structural differences as revealed by the teacher GNN. However, it is important to note that knowledge distillation (KD) establishes statistical correlations between node features and structure, allowing partial structural information to be recovered from node features alone. The extreme case presented, where two nodes have exactly same features but different neighbors, is indeed rare, as in many real-world scenarios, node features are diverse and distinct. For instance, in citation networks, node features are often based on “bag of words” or “word2vec”, ensuring a high degree of diversity among node features. Additionally, even in cases where identical node features exist, techniques such as assigning learnable node embeddings or employing network embeddings (e.g., deepwalk, node2vec) as node features can be applied to mitigate this issue.
>
> [1] Tian Y, Zhang C, Guo Z, et al. Learning mlps on graphs: A unified view of effectiveness, robustness, and efficiency[C]//The Eleventh International Conference on Learning Representations. 2023.

---

> ### Author Response · Authors · 2023-11-20
> **Gentle Reminder**
>
> Dear Esteemed Reviewer 12WL,
>
>
> We sincerely appreciate the time and effort you dedicated to reviewing our paper. Your thoughtful questions and insightful feedback have been extremely beneficial. In response to your queries and concerns, we have prepared detailed answers .
>
> We understand that you have numerous commitments, and we truly appreciate the time you invest in our work. As the discussion period is approaching its conclusion in one day, we kindly request, if possible, that you review our rebuttal at your earliest convenience.
>
> Should there be any further points that require clarification or improvement, please know that we are fully committed to addressing them promptly.
>
> Thank you once again for your invaluable contribution to our research.
>
> Warm Regards,
> The Authors

---

> ### Author Response · Authors · 2023-11-23
>
> Dear Reviewer 12WL:
>
> We just want to reach out to you again and see if our response addresses your concern. Your comments really inspire us, and we are eager to continue discussing our work with you.
>
> Warm Regards,
>
> The Authors

---

> > ### Comment · Reviewer_12WL · 2023-11-23
> > **Thank you! Comment by Reviewer**
> >
> > Thank you for further clarification! So, according to the paper and the response, my understanding is that the distillation indeed has a problem that the structural difference of nodes may not be learned by the student MLP, and additional tricks like feature augmentation have to be used in practice. This would somewhat reduce the contribution of the proposed approach claimed by the paper (but I understand previous work used the same setting)
> >
> > I will keep my rating, but I won't object if all the other reviewers recommend acceptance.

---

> ### Author Response · Authors · 2023-11-23
> **Response to Reviewer 12WL**
>
> Thanks for your response. In this paper, regarding GNN-to-MLP distillation, we strictly follow the task and experimental settings of GLNN [1] and NOSMOG [2] (ICLR 2023 notable top 25%) for fair comparisons, and we outperform them from different aspects. Besides, a more impactful contribution of our method is we find a new effective GNN representation space (i.e., the codebook) for maximally preserving diverse local graph structures, which may **not only enhance GNN-to-MLP distillation, but also benefit other GNN-based applications, such as heterophilic graphs** (as illustrated in the responses to other reviewers). We believe this would contribute to the community and we will continue to explore its new potentials. Finally, we still highly appreciate your time and effort in reviewing our paper.
>
>
> [1] Zhang S, Liu Y, Sun Y, et al. Graph-less Neural Networks: Teaching Old MLPs New Tricks Via Distillation[C]//International Conference on Learning Representations. 2022.
>
> [2] Tian Y, Zhang C, Guo Z, et al. Learning mlps on graphs: A unified view of effectiveness, robustness, and efficiency[C]//The Eleventh International Conference on Learning Representations. 2023.
>
> Warm Regards,
>
> The Authors

---

### Author Response · Authors · 2023-11-17
**General Response**

We sincerely thank all the reviewers for the thorough reviews and valuable feedback. We are glad to hear that the idea is novel (all reviewers), the paper is well-written and easy to follow (Reviewer CnQf, F8Tt, 4k7i, and T74P), the experiments are comprehensive and convincing (Reviewer Cnqf, F8Tt and 4K7i), and the approach is effective on different datasets and tasks (Reviewer F8Tt, 4K7i, T74P). We have revised the manuscript according to the suggestions of reviewers (**mainly in the appendix part, marked in red**).

We here summarize and highlight our responses to the reviewers:

* We apply our proposed method on heterophilic graphs (Reviewer Cnqf, 4K7i) and provide experimental results in the updated manuscript, further demonstrating the generalization ability of our Graph VQ-VAE.

* We provide detailed explanations regarding choice about codebook size (Review F8Tt, 4K7i) and the differences with other representation learning methods (Reviwer 4K7i, T74P).

* We illustrate further connections between GNN-to-MLP distillation and a local structure-aware codebook (Reviewer Cnqf), a theoretical understanding of our method (Reviewer F8Tt).

We reply to each reviewer's concerns in detail below their reviews. Please kindly check out them. Thank you and please feel free to ask any further question.

---

### Meta-Review · Area_Chair_MWnw · 2023-12-13

**Metareview:**

This paper investigates the GNN-to-MLP distillation problem for node classification.  It claims that the output representation of teacher GNN lacks rich graph structural information, and then proposes VQGraph, as a variation of VQ-VAE.  VQGraph incorporates a VQ-VAE to learn a codebook that represents informative local structures, and uses these local structures as additional information for distillation. Empirically, this approach outperforms state-of-the-art methods, in both accuracy and inference speed.

The proposed VQGraph seems plausible, and the paper is well written, with comprehensive and convincing experimental results. The paper will further benefit from a theoretical understanding of the proposed approach.

**Justification For Why Not Higher Score:**

The paper will further benefit from a theoretical understanding of the proposed approach.

**Justification For Why Not Lower Score:**

The proposed VQGraph seems plausible, and the paper is well written, with comprehensive and convincing experimental results.

---

### Decision · Program_Chairs · 2024-01-16

Accept (poster)